# Mutant Kras- and p16-regulated NOX4 activation overcomes metabolic checkpoints in development of pancreatic ductal adenocarcinoma

Huai-Qiang Ju[1,2], Haoqiang Ying[2], Tian Tian[1], Jianhua Ling[2], Jie Fu[2], Yu Lu[2], Min Wu[2], Lifeng Yang[3], Abhinav Achreja[3], Gang Chen[4], Zhuonan Zhuang[2], Huamin Wang[5], Deepak Nagrath[3], Jun Yao[2], Mien-Chie Hung[2,6], Ronald A. DePinho[6,7], Peng Huang[1,4,6], Rui-Hua Xu[1] & Paul J. Chiao[2,6]

Kras activation and p16 inactivation are required to develop pancreatic ductal adenocarcinoma (PDAC). However, the biochemical mechanisms underlying these double alterations remain unclear. Here we discover that NAD(P)H oxidase 4 (NOX4), an enzyme known to catalyse the oxidation of NAD(P)H, is upregulated when p16 is inactivated by looking at gene expression profiling studies. Activation of NOX4 requires catalytic subunit p22[phox], which is upregulated following Kras activation. Both alterations are also detectable in PDAC cell lines and patient specimens. Furthermore, we show that elevated NOX4 activity accelerates oxidation of NADH and supports increased glycolysis by generating $NAD^+$, a substrate for GAPDH-mediated glycolytic reaction, promoting PDAC cell growth. Mechanistically, NOX4 was induced through p16-Rb-regulated E2F and p22[phox] was induced by $Kras^{G12V}$-activated NF-κB. In conclusion, we provide a biochemical explanation for the cooperation between p16 inactivation and Kras activation in PDAC development and suggest that NOX4 is a potential therapeutic target for PDAC.

[1] Sun Yat-sen University Cancer Center, State Key Laboratory of Oncology in South China, Collaborative Innovation Center for Cancer Medicine, Guangzhou 510060, China. [2] Department of Molecular and Cellular Oncology, The University of Texas MD Anderson Cancer Center, Houston, Texas 77030, USA. [3] Laboratory for Systems Biology of Human Diseases, Rice University, Houston, Texas 77005, USA. [4] Department of Translational Molecular Pathology, The University of Texas MD Anderson Cancer Center, Houston, Texas 77030, USA. [5] Department of Pathology, The University of Texas MD Anderson Cancer Center, Houston, Texas 77030, USA. [6] The University of Texas Graduate School of Biomedical Sciences, Houston, Texas 77030, USA. [7] Department of Cancer Biology, The University of Texas MD Anderson Cancer Center, Houston, Texas 77030, USA. Correspondence and requests for materials should be addressed to P.H. (phuang@mdanderson.org) or to R.-H.X. (email: xurh@sysucc.org.cn) or to P.J.C. (email: pjchiao@mdanderson.org).

The mutational activation of Kras is an early genetic alteration in the development of pancreatic ductal adenocarcinoma (PDAC). This alteration has been detected in nearly 95% of PDAC cases, and mutational inactivation of the $p16^{INK4a}$ (hereafter referred to as p16) tumour suppressor gene can be identified in $\sim$80–90% of PDAC cases[1–3]. Previous studies showed that $Kras^{G12D}$ played critical roles in initiating and maintaining PDAC, however, activation of Kras alone may not be sufficient to initiate tumorigenesis[4,5]. The mutant Kras mouse models have demonstrated that additional inactivation of $p16/p14$ or $p53$ dramatically accelerated the progression of $Kras^{G12D}$ initiated PDAC[6,7]. Recent studies by our group and others showed that $Kras^{G12V}$ activation led to suppression of mitochondrial respiratory activity and rendered the cell more dependent on glycolysis[6,8,9]. Conversely, others reported that mitochondrial reactive oxygen species (ROS) generation is essential for $Kras^{G12V}$-induced cell proliferation and $Kras^{G12V}$-mediated tumorigenicity[10]. Dysfunctional mitochondria and increased aerobic glycolysis are two important biochemical characteristics observed frequently in cancer cells[8,11,12]. A metabolic switch from oxidative phosphorylation in the mitochondria to glycolysis in the cytosol in cancer cells has been well known as 'Warburg effect' for decades[13,14]. Although reprogramming of cellular metabolism is now recognized as a key event during tumorigenesis, the molecular mechanisms that initiate this metabolic shift in tumorigenesis remain elusive. Also, the Ras downstream effectors in tumorigenesis are elusive, the mechanisms by which inactivated p16 cooperates with oncogenic Kras to bypass the metabolic checkpoint through activation of specific signalling pathways essential for $Kras^{G12V}$-mediated tumorigenesis remain unclear.

The family of NAD(P)H oxidases (NOX) consists of five members (NOX1-5) and the small membrane-bound catalytic subunit ($p22^{phox}$) required for their activation. As ROS-generating enzymes, all the NOX members have the capacity to oxidize the substrates NADPH or NADH to $NADP^+$ or $NAD^+$, which in turn results in formation of superoxide[15,16]. Previous studies showed that NOX oxidases can effect several of the hallmarks of cancer, including genomic instability, autonomous growth and survival, invasion and metastasis, possibly because of the alteration of redox-linked signalling systems that are influenced by NOX-derived ROS[17]. However, the biochemical roles for specific NOX complexes in cancer that are relevant to cellular metabolism and the 'Warburg effect' remain unexplored.

We previously demonstrated that the expression of mutant Kras and the silencing of $Kras^{G12V}$-induced p16 expression in hTERT-immortalized human pancreatic nestin-positive epithelial (HPNE) cells resulted in transformation in vitro and in development of PDAC in vivo[5], Overexpression of NOX4 was identified in HPNE/$Kras^{G12V}$/shp16 cells, verified in PDAC cells and patient specimens. NOX4 has been reported to play an important pro-survival role in pancreatic cancer via unclear mechanisms[18,19]. However, the function and molecular mechanisms of NOX4 activation in reprogramming the metabolism, tumorigenic transformation and pancreatic carcinogenesis remained unknown.

In the present study, we investigate the function of NOX4 in $Kras^{G12V}$ activation- and p16 inactivation-induced PDAC and explored the underlying regulatory mechanisms. We show that NOX4 activity is activated by increased expression of both NOX4 by p16-Rb-E2F and $p22^{phox}$ via $Kras^{G12V}$-NF-κB pathways to overcome metabolic checkpoints to enable initiation of PDAC development. Our findings suggest that NOX4 is a potential therapeutic target for cancer therapy.

## Results

**Mutant Kras and p16 increase NOX4 activity.** To study the mechanisms of tumorigenesis in the pancreas, we established an HPNE cell model expressing $Kras^{G12V}$ or $Kras^{G12V}$/p16shRNA[5]. We found that activation of $Kras^{G12V}$ alone, which was not sufficient to initiate tumorigenesis in HPNE cells, induced high expression of p16 (Fig. 1a). It is well known and we confirmed that loss of p16 is the most common mutation in PDAC cells, PanIN (pancreatic intraepithelial neoplasia) and PDAC tissues (Supplementary Fig. 1a,b). Interestingly, silencing p16 expression in $Kras^{G12V}$ cells resulted in tumorigenic transformation and development of PDAC in an orthotopic xenograft mouse model[5]. The analysis of Kras copy number indicates the ratio between the HPNE/$Kras^{G12V}$ and HPNE cells is about 4 times (Fig. 1a), which is consistent with the recent finding that mutant Kras copy gains are positively selected during tumour progression in KPC lung cancer mouse model[20,21]. To elucidate the downstream pathways activated by oncogenic Kras and inactivated p16 in human pancreatic tumorigenesis, we profiled gene expression in HPNE/$Kras^{G12V}$/shp16 and HPNE/$Kras^{G12V}$ cells using cDNA microarray analysis (Fig. 1b). Bioinformatics analysis identified 614 genes whose expression was significantly increased on p16 knockdown. Figure 1c shows the functional categories of the upregulated genes as predicted by gene set enrichment and pathway analyses. It indicated that the most elevated genes in tumorigenic HPNE/$Kras^{G12V}$/shp16 cells were associated with metabolic processes. NOX4, a key enzyme known to catalyse the oxidation of NADPH or NADH to $NADP^+$ or $NAD^+$, was the only metabolic enzyme among the top ten highly expressed genes in response to p16 knockdown in our microarray (Fig. 1b; Supplementary Table 1). Oncogenic Kras was shown to alter metabolism, but how mutant Kras induces metabolic reprogramming that contributes to tumorigenic transformation is unknown. To illuminate the mechanistic links between activated Kras, inactivated p16 and overexpressed NOX4 in regulation of metabolism, we investigated whether and how energy metabolism was regulated by NOX4, and how oncogenic Kras cooperates with inactivated p16 to increases the expression and activity of NOX4.

To verify the expression of NOX4 and its catalytic subunit $p22^{phox}$ in HPNE, HPNE/$Kras^{G12V}$ and HPNE/$Kras^{G12V}$/shp16 cells, we performed qPCR and immunoblotting analysis. As shown in Fig. 1d, $p22^{phox}$ expression was induced by $Kras^{G12V}$ in both HPNE/$Kras^{G12V}$ and HPNE/$Kras^{G12V}$/shp16 cells, while NOX4 was induced only in HPNE/$Kras^{G12V}$/shp16 cells at both the mRNA and protein levels. Further analysis also revealed similar results after depletion of p16 using two different siRNAs in HPNE/$Kras^{G12V}$ cells (Fig. 1h). Moreover, the activation of Kras in HPNE cells resulted in a moderate increase of NOX activity, and silencing of p16 in HPNE/$Kras^{G12V}$ cells led a further increase in NOX activity (Fig. 1e). Consistent with NOX as a major source of ROS[17], HPNE/$Kras^{G12V}$ and HPNE/$Kras^{G12V}$/shp16 cells showed a substantial increase in superoxide ($O_2^-$) levels. In response to NOX-induced ROS stress, the cellular glutathione (GSH) and GSH/GSSG ratio was significantly increased in HPNE/$Kras^{G12V}$ and HPNE/$Kras^{G12V}$/shp16 cells (Supplementary Fig. 1c,d). Taken together, these data suggested that activation of Kras with silencing of p16 led to NOX-induced ROS generation and a compensatory increase in cellular antioxidant activity. To further verify these above findings, we examined the expression and activities of NOX4 and $p22^{phox}$ in human pancreatic ductal epithelial (HPDE)/$Kras^{G12V}$ and HPDE/$Kras^{G12V}$/shp16 cells derived from the nontumorigenic immortalized HPDE cells[22]. Consistent with our observations in the HPNE cell models, NOX4 expression, NOX activity and basal $O_2^-$ levels were significantly elevated in HPDE/$Kras^{G12V}$ and HPDE/$Kras^{G12V}$/shp16 cells than in parental HPDE cells

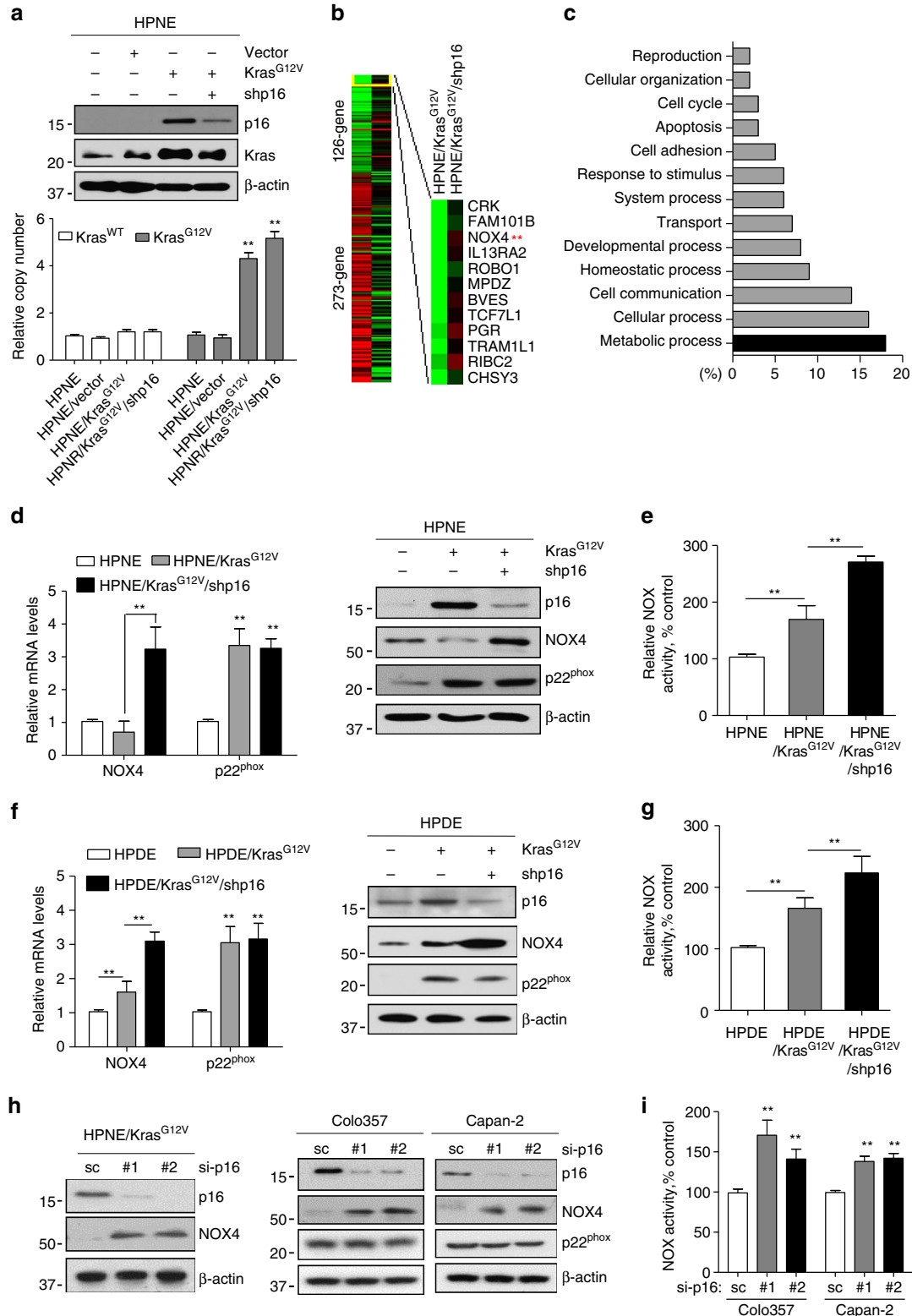

**Figure 1 | Activated Kras or silenced p16 increased NOX4/p22$^{phox}$ expression and elevated NOX activity.** (**a**) The expression of Kras and p16 was analysed by immunoblotting in HPNE control, HPNE/Kras and HPNE/Kras$^{G12V}$/shp16 cells. The copy numbers of Kras$^{WT}$ and Kras$^{G12V}$ were analysed by qPCR assay in these cells. (**b**) Heat map of gene expression between HPNE/Kras and HPNE/Kras$^{G12V}$/shp16 cells identified using cDNA microarray. (**c**) Functional categorization of 614 genes upregulated in response to p16 suppression in HPNE/Kras$^{G12V}$ cells. (**d,f**) The expression of NOX4 and p22$^{phox}$ was analysed by qPCR and immunoblotting in indicated cells. (**h**) The expression of NOX4, p22$^{phox}$ and p16 was analysed by immunoblotting in HPNE/Kras$^{G12V}$, Colo357 and Capan-2 cells transfected with two independent p16 siRNAs. (**e,g,i**) NOX activity was determined in indicated cells by measuring NADPH-dependent superoxide ($O_2^-$) generation with the lucigenin-enhanced chemiluminescence assay. β-actin was used as the internal loading control. Each bar represents the mean ± s.d. Data in **d,e,g,i** are presented as mean ± s.d. ($n = 3$). **$P < 0.01$ for indicated comparison (one-way analysis of variance (ANOVA) with the Newman Keul's multiple comparison test).

(Fig. 1f,g; Supplementary Fig. 1e). Further analysis revealed that the expression of NOX4 level was also increased after siRNA depletion of p16 in Colo357 and Capan-2 cells with wild-type p16, however the expression of p22phox level was not changed on p16 depletion (Fig. 1h). NOX activity and cellular $O_2^-$ levels were also significantly increased after siRNA depletion of p16 in these cells (Fig. 1g,i; Supplementary Fig. 1f). Altogether, these data suggest that either activation of Kras or inactivation of p16 induces NOX activation by increasing the expression of NOX4 or p22phox in HPNE, HPDE and PDAC cells.

**NOX4 and p22phox are overexpressed in PDAC.** To determine the role of NOX in pancreatic tumorigenesis, we examined the pancreas-specific expression of the five NOX family members and the catalytic subunit p22phox in HPNE cells and 11 indicated PDAC cell lines. RT-PCR analysis showed that both NOX4 and p22phox mRNAs were expressed in HPNE and all the tested PDAC cells (Supplementary Fig. 2a,b). NOX2 and NOX5 could be detected in only few cells, whereas NOX1 and NOX3 expression were not detectable in these cell lines (Supplementary Fig. 2b). The mRNA levels of NOX4 and p22phox were significantly increased in these PDAC cells or PDAC tissues (Fig. 2a,b). Furthermore, immunoblotting analysis showed that the NOX4 and p22phox protein levels were substantially higher in PDAC cells than in nontumorigenic HPNE cells (Fig. 2c). Consistently, the NOX activity were significantly increased in these PDAC cells (Fig. 2d). Thus, the expression levels of NOX4 and p22phox were overexpressed in PDAC cells and human PDAC tissues.

To determine whether the expression of NOX4 was increased in genetically engineered mouse models (GEMM) from *Pdx1-Cre*; *Kras^{G12D}*; *p16^{F/F}* mice, immunohistochemical (IHC) analyses were performed. NOX4 levels were substantially higher in these tumours than in histologically normal pancreata from control mice (Fig. 2e). Furthermore, IHC staining indicated that NOX4 levels were also elevated in PDAC tumours from *iKras*; *p53^{L/+}* mouse model than in histologically normal pancreata from control mice (Supplementary Fig. 2c). Thus, the expression levels of NOX4 were increased in PDAC. Prior study showed that p22phox levels were significantly higher in pancreatic carcinoma than in non-malignant tissues[23]. To further determine the clinical relevance of NOX4 expression, we then analysed the expression of NOX4 in a pancreatic tissue microarray representing 120 cases of PDAC and 110 normal pancreatic acinous tissues. As shown in Fig. 2f, 81.7% (98/120) of the pancreatic cancer tissues exhibited high levels of NOX4, whereas only about 35.5% (39/110) of the pancreatic tissues with normal histological appearance were positive for NOX4 expression. Statistical analysis showed that the increase in NOX4 expression in PDAC patients is highly significant ($P < 0.001$) (Fig. 2f). Furthermore, the increased expression of NOX4 and p22phox in PDAC analysed from multiple cancer microarray data sets available from Oncomine also supports our findings (Fig. 2g). Altogether, the data from gene and protein profiling, cell lines and human and mice PDAC tissues suggest that NOX4 expression is significantly increased in PDAC and NOX4 may play a key role in PDAC tumorigenesis.

**NOX4 supports increased glycolysis by generating NAD$^+$.** Cancer cells abnormally take up more glucose, processing through aerobic glycolysis, then produces high levels of secreted lactate, this phenomenon is called the 'Warburg effect', which was interpreted as mitochondrial dysfunction[14,24]. Indeed, the mitochondrial respiratory chain activity was reduced in HPNE/Kras^{G12V} and HPNE/Kras^{G12V}/shp16 cells as evinced by a substantial decrease in oxygen consumption rate (OCR)

(Fig. 3a). Accordingly, we found that glucose uptake and lactate production levels were significantly increased in HPNE/Kras^{G12V}/shp16 cells than in HPNE/Kras^{G12V} and HPNE cells (Fig. 3b). In the glycolytic pathway, glyceraldehyde-3-phosphate dehydrogenase (GAPDH) catalyses the conversion of glyceraldehyde-3-phosphate to D-glycerate 1,3-bisphosphate and NAD$^+$ serving as a coenzyme is simultaneously reduced to NADH[25]. NAD$^+$ is mainly regenerating from NADH by lactate dehydrogenase isoform A (LDHA), which preferentially converts accumulating pyruvate to lactate, to maintain glycolytic flux in cancer cells[25]. On the basis of the LDHA reaction, the accumulation of lactate in tumour implies an increase of NADH relative to NAD$^+$: [Lactate]/[Pyruvate] $\propto$ [NADH][NAD$^+$], which means the NAD$^+$ is insufficient in cancer cells with higher glycolytic activity (Supplementary Fig. 3a)[26]. Also, the negative feedback regulation of phosphofructokinase 1 (PFK1) by lactate act as strong inhibitors is overridden by the allosteric activator, fructose-2,6-bisphosphate (F2,6BP)[27]. Our further results also confirmed that the cellular NAD$^+$/NADH and NADP$^+$/NADPH was significantly decreased in HPNE/Kras^{G12V}/shp16 cells with higher lactate concentration (Fig. 3b). Besides, the higher levels of NADPH may produce by other metabolism pathways to counteract oxidative stress and to support biomass production[28]. Because NOX4 has the capacity to oxidize the substrates NADPH or NADH to NADP$^+$ or NAD$^+$ (ref. 15), we posited that NOX4 may function as a compensatory mechanism and accelerates NAD$^+$ and NADP$^+$ circulation to sustain cellular glycolysis and pentose phosphate pathway activity in mitochondria-defective pancreatic cancer cells.

To further substantiate the essentiality of NOX4 in NAD$^+$ and NADP$^+$ generation and glucose metabolism, we evaluated the effect of NOX4 knockdown on cellular NAD$^+$, NADP$^+$ and glycolytic activity. We measured cellular NADH and NAD$^+$ contents in HPNE/Kras^{G12V}/shp16 cells in the presence and absence of NOX4 siRNA. Inhibition of NOX activity by NOX4 siRNA knockdown caused significant decreases in cellular NAD$^+$/NADH and NADP$^+$/NADPH ratios, by ∼30 and 20%, respectively (Fig. 3c,d). Furthermore, knockdown of NOX4 also significantly decreased glucose uptake, lactate production and ATP generation (Fig. 3d), indicating that NOX4 has an important function in glycolytic metabolism of PDAC cells. Glycolytic activity also was decreased in HPNE/Kras^{G12V}/shp16, AsPc-1 and Panc-28 cells with silenced p22phox or NOX4 (Fig. 3c,e,f). Also, the decreased NAD$^+$/NADH, NADP$^+$/NADPH ratios and glycolytic activity were significantly rescued in these cells co-transfected with human NOX4 siRNA and siRNA resistance NOX4 (NOX4-R) (Fig. 3f; Supplementary Fig. 3b,c). To strengthen our experimental evidence, we established NOX4-overexpressing HPNE/NOX4 cells and performed metabolite isotope tracing experiments with 13 carbon labelled glucose (U-$^{13}C_6$ Glu). We found that overexpression of NOX4 in HPNE/NOX4 and HPNE/Kras^{G12V}/shp16 in HPNE cells increased pyruvate and lactate levels, thereby confirming the increased glucose to lactate conversion or glycolysis (Fig. 3g,h). These findings unequivocally demonstrate that NOX4 plays an essential role in maintaining high glycolytic activity in PDAC cells by supplying NAD$^+$ from oxidation of NADH.

**Expression of p22phox is upregulated by Kras-NF-κB pathway.** To interrogate the molecular mechanism through which p22phox overexpression was induced by oncogenic Kras in PDAC cells, we investigated whether inhibitors of the major Kras downstream signalling pathways, including PI3K/Akt, ERK1/2 and NF-κB, would decrease p22phox expression. We found that the expression

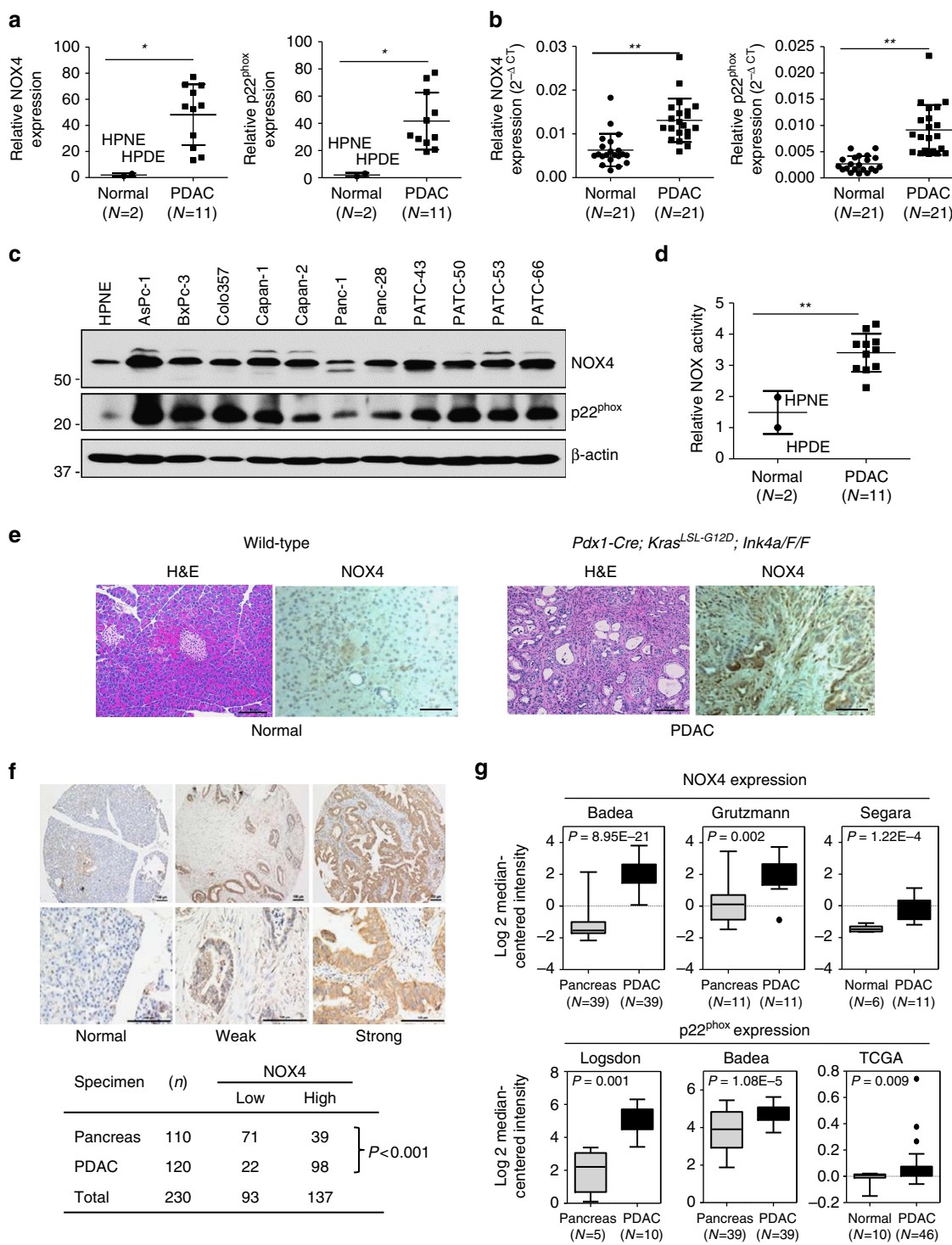

**Figure 2 | NOX4 and p22^phox are overexpressed in PDAC. (a)** The mRNA levels of NOX4 and p22^phox were compared between 11 PDAC cell lines and 2 normal cell lines. (**b**) The mRNA levels of NOX4 and p22^phox were compared between PDAC tissues and adjacent normal tissues ($N = 21$). (**c**) The expression of NOX4 and p22^phox in different PDAC cell lines was analysed by immunoblotting and compared with that in HPNE cells. β-actin was included as a loading control. (**d**) NOX activity was compared between 11 PDAC cell lines and 2 normal cell lines. (**e**) Representative IHC staining with H&E or anti-NOX4 antibody in sections of formalin-fixed tissue from wild-type or *Pdx1-Cre; Kras^LSL-G12D*, *Ink4a^F/F* mice. (**f**) Representative IHC staining showing no expression of NOX4 in normal pancreas acinous cells, weak and strong positive staining ($\times 10$) in PDAC tissues. Lower panels represent higher magnifications ($\times 40$). NOX4 expression is considered to be significantly different between PDAC and normal tissues and higher in PDAC group ($P < 0.0001$ analysed by Fisher's exact test). Scale bars in **e**, **f**, 100 μm. (**g**) NOX4 and p22^phox expression in multiple cancer microarray data sets available from Oncomine (https://www.oncomine.com//). Data in **a,b,d** are representative of three independent experiments and presented as mean ± s.d. *$P < 0.05$, **$P < 0.01$ for indicated comparison (Student unpaired *t*-test). In **g**, box plot centre line and box limits represent median and interquartile range (IQR), respectively. Whisker lines represent data range (maximum and minimum) but not exceeding 1.5 × IQR from IQR. Data points beyond 1.5 × IQR from IQR are shown with circles.

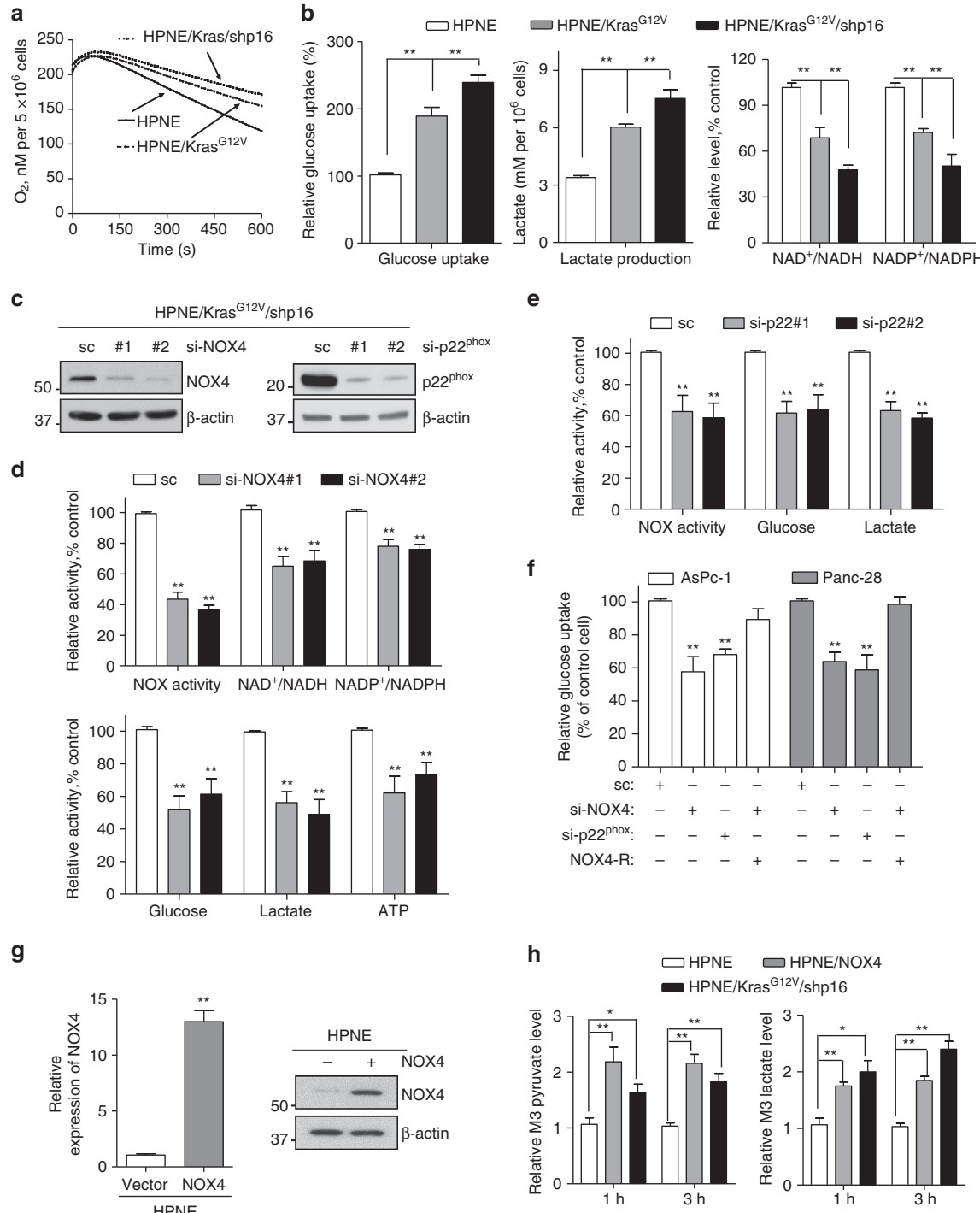

**Figure 3 | NOX4 plays a critical role in regulation of glycolysis by generating NAD+.** (**a**) OCR was determined in HPNE, HPNE/KrasG12V and HPNE/KrasG12V/shp16 cells. (**b**) Glucose uptake and lactate production, NAD+/NADH and NADP+/NADPH ratios were measured in HPNE, HPNE/KrasG12V and HPNE/KrasG12V/shp16 cells. (**c**) Immunoblotting analysis showed the NOX4 and p22phox knockdown efficiency. Scrambled siRNA (sc) was used as a negative control here and in **d**–**f**. (**d**) NOX activity, NAD+/NADH, NADP+/NADPH ratios and glycolytic activity (glucose uptake, lactate production and ATP levels) were measured in NOX4-silenced HPNE/KrasG12V/shp16 cells. (**e**) NOX activity, glucose uptake and lactate production levels were measured in p22phox-silenced HPNE/KrasG12V/shp16 cells. (**f**) Glucose uptake was measured in AsPc-1 and Panc-28 cells after siRNA depletion of NOX4 or p22phox or co-expressing siRNA resistant NOX4 (NOX4-R). (**g**) The expression levels of NOX4 were analysed by qPCR and immunoblotting in NOX4-overexpressing HPNE cells. Data are presented as mean ± s.d. ($n = 3$). **$P < 0.01$ for indicated comparison (Student unpaired $t$-test). (**h**) The pyruvate and lactate levels were measured in HPNE/NOX4 or HPNE/KrasG12V/shp16 cells compared with parental HPNE cells using metabolite isotope tracing experiments with 13 carbon labelled glucose (U-13C6 Glu). Data in **b**,**d**–**f**,**h** are presented as mean ± s.d. ($n = 3$). *$P < 0.05$, **$P < 0.01$ for indicated comparison (one-way analysis of variance (ANOVA) with the Newman Keul's multiple comparison test).

of p22[phox] mRNA and protein levels were significantly decreased only in HPNE/Kras[G12V]/shp16, AsPc-1 and Panc-28 cells treated with NF-κB inhibitor PDTC (Supplementary Fig. 4a,b). Furthermore, immunoblotting analysis showed that expression levels of phosphorylated (p)-TAK1 and p-NF-κB (that is, activated NF-κB/p65) were elevated in HPNE/Kras and HPNE/Kras[G12V]/shp16 cells, indicating that the TAK1-NF-κB signalling pathway

was activated (Fig. 4a), which is consistent with our previous published results[29]. Expression of p22[phox] and NOX activity were decreased on induction of doxycycline (Dox)-regulated TAK1 shRNA in AsPc-1/iTAK1shRNA cells (Fig. 4b). Expression of p22[phox] was induced by Dox following NF-κB activation in iKras cells established from a PDAC mouse model with the Dox-inducible Kras[G12V] transgene[7] (Fig. 4c). Silencing of p65/NF-κB

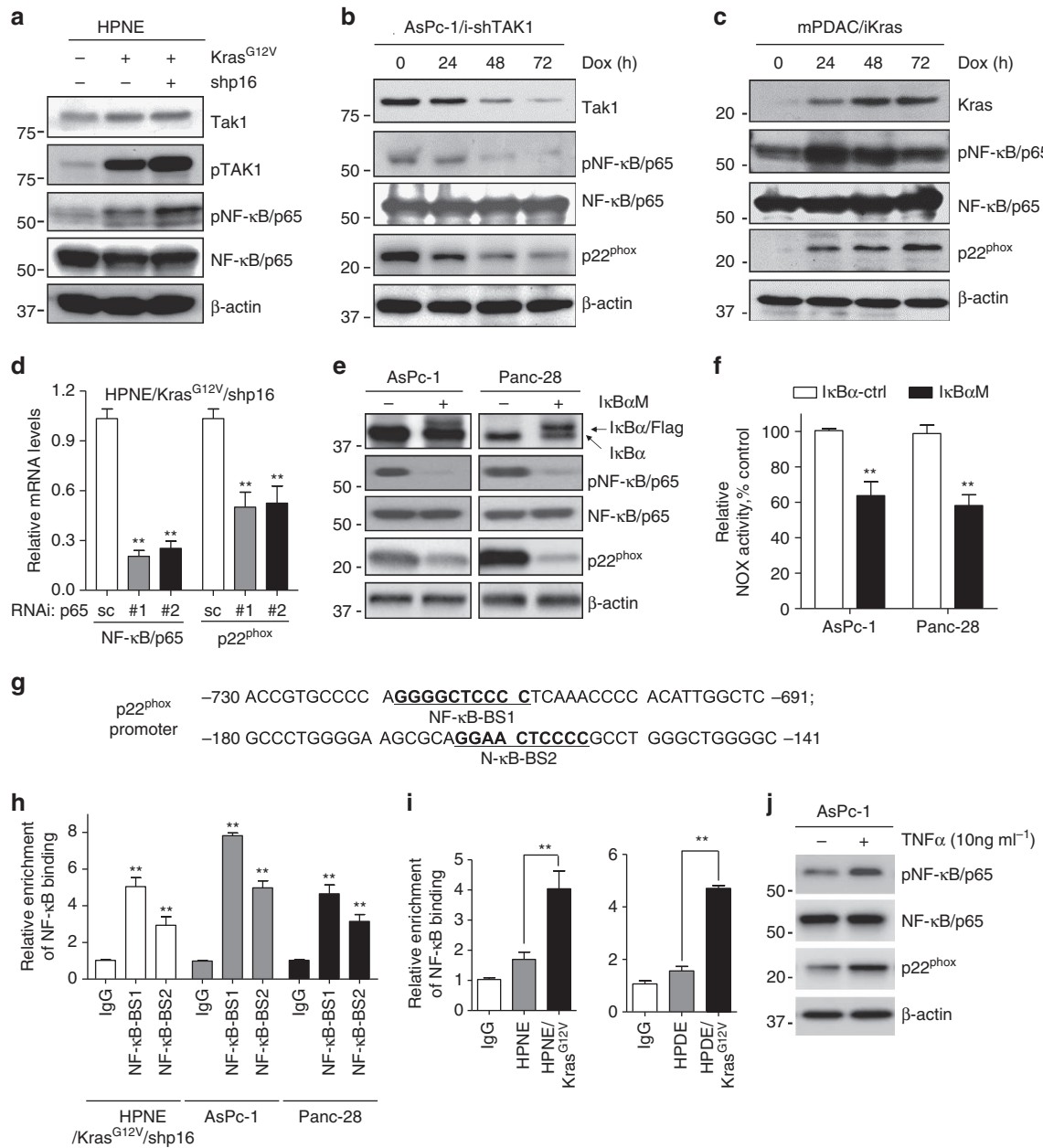

**Figure 4 | Kras upregulates p22[phox] expression via the Tak1-NF-κB pathway.** (**a**) The expression of p-Tak1, Tak1, p-NF-κB/p65 and NF-κB/p65 was analysed by immunoblotting in HPNE, HPNE/Kras[G12V] and HPNE/Kras[G12V]/shp16 cells. (**b**) The expression of Tak1, p-NF-κB/p65, NF-κB/p65 and p22[phox] was analysed by immunoblotting in AsPc-1/i-Tak1shRNA cells with Dox induction. (**c**) The expression of Kras, p-NF-κB/p65, NF-κB/p65 and p22[phox] was analysed by immunoblotting in mPDAC/iKras cells with Dox induction. (**d**) The NF-κB/p65 and p22[phox] mRNA expression was analysed by qPCR in HPNE/Kras[G12V]/shp16 cells transfected with NF-κB/p65 siRNAs. (**e**) The expression of phosphorylation-defective mutant IκBα, p-NF-κB/p65, NF-κB/p65 and p22[phox] was analysed by immunoblotting in wild-type (WT) and IκBα-mutant (Mu) AsPc-1 and Panc-28 cells. (**f**) NOX activity was compared in IκBα wild-type and mutant AsPc-1 and Panc-28 cells. (**g**) The sequences of human p22[phox] promoter regions are presented with NF-κB binding sites (BS-1 and BS-2) indicated. (**h**) The activities of NF-κB binding sites BS-1 and BS-2 in p22[phox] promoter were analysed in indicated cells by ChIP and qPCR assay. IgG was used as a negative control. (**i**) The activities of NF-κB binding sites BS-1 in p22[phox] promoter were analysed in indicated cells by ChIP assay and qPCR. (**j**) The expression of p-NF-κB/p65, NF-κB/p65 and p22[−phox] was analysed by immunoblotting in AsPc-1 cells treated with TNF-α (10 ng ml[−1]) for 48 h. β-actin was used as the internal loading control. Data in **d,f,h,i** are presented as mean ± s.d. ($n = 3$). **$P < 0.01$ for indicated comparison (Student unpaired t-test).

in HPNE/Kras$^{G12V}$/shp16 cells with p65/NF-κB siRNA resulted in downregulation of p22$^{phox}$ mRNA as determined by qPCR analysis (Fig. 4d). In Panc-28 and AsPc-1 cells expressing a mutant of IκBα with a defect in phosphorylation, p22$^{phox}$ expression was substantially inhibited and NOX activity was significantly decreased (Fig. 4e,f). These results suggest that p22$^{phox}$ overexpression was induced by NF-κB. To determine whether p22$^{phox}$ expression is regulated at transcription by NF-κB, we identified two NF-κB binding sites in 3-kb human p22$^{phox}$ promoters (Fig. 4g). Chromatin immunoprecipitation (ChIP) assays revealed that the NF-κB-BS-1 site had higher binding activity in HPNE/Kras$^{G12V}$/shp16, AsPc-1 and Panc-28 cells than in controls (Fig. 4h). The NF-κB-BS-1 site also showed higher binding activity in all the HPNE/Kras and HPDE/Kras cells compared with the parental cells (Fig. 4i). Further study also indicates that the expression of p-NF-κB and p22$^{phox}$ was obviously increased in AsPc-1 cells on TNFα stimulation (Fig. 4j). In addition, we found that expression of NOX4 was induced by Dox in mPDAC/iKras cells, which was consistent with the IHC staining result using the transgenic *iKras* mice tissue. However, NOX4 levels were unchanged in AsPc-1/iTAK1shRNA cells, and Panc-28 or AsPc-1 cells expressing a mutant of IκBα (Supplementary Fig. 4c–e). These findings demonstrate that NF-κB is a major transcription factors binding to p22$^{phox}$ promoter and mutant Kras-activated NF-κB mainly regulates p22$^{phox}$ expression in PDAC cells.

**NOX4 is upregulated via the p16-Rb-E2F pathway**. The p16 pathway regulates cell cycle entry and progression, and loss of p16 activates *CDK4*, which in turn phosphorylates retinoblastoma protein (Rb), causing it to release the transcription factor E2F (ref. 30). To further dissect whether loss of p16 upregulates NOX4 expression via the Rb-E2F pathway in pancreatic cancer cells, we performed an immunoblotting analysis, which showed that Rb was strongly phosphorylated at sites S795 and S807/811 and that the expression level of nuclear E2F1 was increased in HPNE/Kras$^{G12V}$/shp16 cells (Fig. 5a). To determine the expression profiles of E2F family members in pancreatic cancer, RT-PCR analyses were performed and revealed that *E2F1*, *E2F3*, *E2F4* and *E2F5* were all expressed in HPNE/Kras$^{G12V}$/shp16, AsPc-1 and Panc-28 cells, while *E2F2* was detected only in AsPc-1 cells (Fig. 5b). *E2F1* was expressed at the highest levels of all the E2F family in these cells (Fig. 5b). RNAi depletion of *E2F1*, *E2F3 E2F4* and *E2F5* all resulted in downregulation of NOX4 expression in HPNE/Kras$^{G12V}$/shp16 and Panc-28 cells (Fig. 5c), demonstrating that NOX4 expression is regulated by several E2F family members. To demonstrate the regulation of NOX4 expression by *E2F1*, we identified two *E2F1* binding sites in 3-kb human NOX4 promoters (Fig. 5d). ChIP assays revealed that the E2F1-BS-1 site showed higher binding activity in HPNE/Kras$^{G12V}$/shp16, AsPc-1 and Panc-28 cells (Fig. 5e). We also found that NOX activity was significantly decreased after RNAi depletion of *E2F1* in these cells (Fig. 5f). However, further study revealed that the E2F binding activity showed no obvious differences between the p16 wild-type and mutant cells (Fig. 5g), consistence with the expression of NOX4 in these cells (Fig. 2c). Considering that gene transcription is a complicated process, it is likely that other transcription factors, co-activators or other cell cycle inhibitors beyond p16 are involved in the regulation of NOX4. As known, *p53*-regulated *p21* expression is decreased when *p53* is knocked out. As the result, CDK4 is activated and phosphorylates Rb, thus, leading to activation of E2F. Consistently, IHC staining indicated that the expression of NOX4 was substantially higher in these PDAC from *iKras*; *p53*$^{L/+}$ mouse model as described above. Together, although the E2F binding activity and the expression of NOX4 is

complicated, our results revealed that loss of p16 upregulates NOX4 expression through the Rb-E2F signalling pathway in PDAC cells.

**NOX4 is a potential therapeutic target for PDAC**. To further substantiate whether inhibition of NOX4 expression would reverse the tumorigenic phenotype induced by mutant Kras and p16, we stably knocked down NOX4 expression in HPNE/Kras$^{G12V}$/shp16 cells using lentiviral NOX4 shRNAs (Supplementary Fig. 5a). The depletion of NOX4 led to a significant decreases in NOX activity, cell growth and colony formation of HPNE/Kras$^{G12V}$/shp16 cells, and significantly decreased their tumorigenic potential in orthotopic xenograft mice (Fig. 6a–c; Supplementary Fig. 5a). Next, we investigated the effect of NOX4 on the kinetics of PDAC tumour progression by subcutaneous injection. Similarly, we observed that the tumours formed by NOX4-silenced cells grew at a much slower rate than control tumours (Fig. 6d–f). As expected, we also observed that knockdown of NOX4 significantly decreased the glucose uptake and lactate production levels, but increased the OCR in the primary culture cells derived from subcutaneous xenograft mice tumour tissue (Supplementary Fig. 5b). To test the role of NOX4 in PDAC cell survival, we established stably knocked down NOX4 in AsPC-1 cells (Supplementary Fig. 5c). Suppression of NOX4 expression also significantly decreased NOX activity, NAD$^+$/NADH ratio, glucose uptake, lactate generation and led to a significant decrease in cell growth (Supplementary Fig. 5e,f). Inhibition of NOX4 by the inhibitor DPI in AsPc-1 and Panc-28 cells also reduced cell viability with minimal toxicity to HPDE and HPNE cells according to MTS assay (Supplementary Fig. 6a,b), suggesting that DPI may selectively cause death of PDAC cells. Importantly, we found that suppression of NOX4 by shRNA or the inhibitor DPI significantly suppressed PDAC growth *in vivo* (Fig. 6g). IHC staining showed suppression of NOX4 by shRNA or the inhibitor DPI also displayed lower cell proliferation indices (Ki67-positive) and higher cell apoptosis (TUNEL-positive) compared with control tumours (Fig. 6h). We also established the inducible NOX4-konckdown cell line (AsPc-1/i-shNOX4), and revealed that 'Dox/on' group mice showed a significant reduction in tumour burden compared with the control group after 4 weeks of dox treatment (Fig. 6i; Supplementary Fig. 5d). As such, we further assessed the impact of DPI on transgenetic *iKras*; *p53*$^{L/+}$ mice generated from crossing iKras and conditional *p53* knockout (*p53*$^L$) alleles[7]. As shown in Fig. 6j, following Dox treatment at 3 weeks of age, all *iKras*; *p53*$^{L/+}$ mice succumbed to PDAC between 14 and 18 weeks of age (median survival, 16 weeks), whereas the DPI-treated group prolonged these mice survival with a median survival of 20 weeks (Fig. 6j). We analysed the proliferative index of the dissected tumours and found that the proliferation index has rebounded in tumours from the DPI-treated group after treatment for 16 weeks. That may explain why DPI treatment just showed moderate effect (Supplementary Fig. 6c,d). These findings demonstrate that NOX4 is essential for PDAC growth by accelerating NAD$^+$ circulation thus maintaining the high levels of glycolysis activity required for PDAC development driven by Kras and loss of p16. Our results revealed that NOX4 is required for PDAC development and pharmacologic targeting of NOX4 represents a potential novel therapeutic approach for pancreatic cancer.

**Discussion**

That mutational activation of Kras and inactivation of p16 are two signature genetic alterations required for development of PDAC, and it has been demonstrated in mouse models that

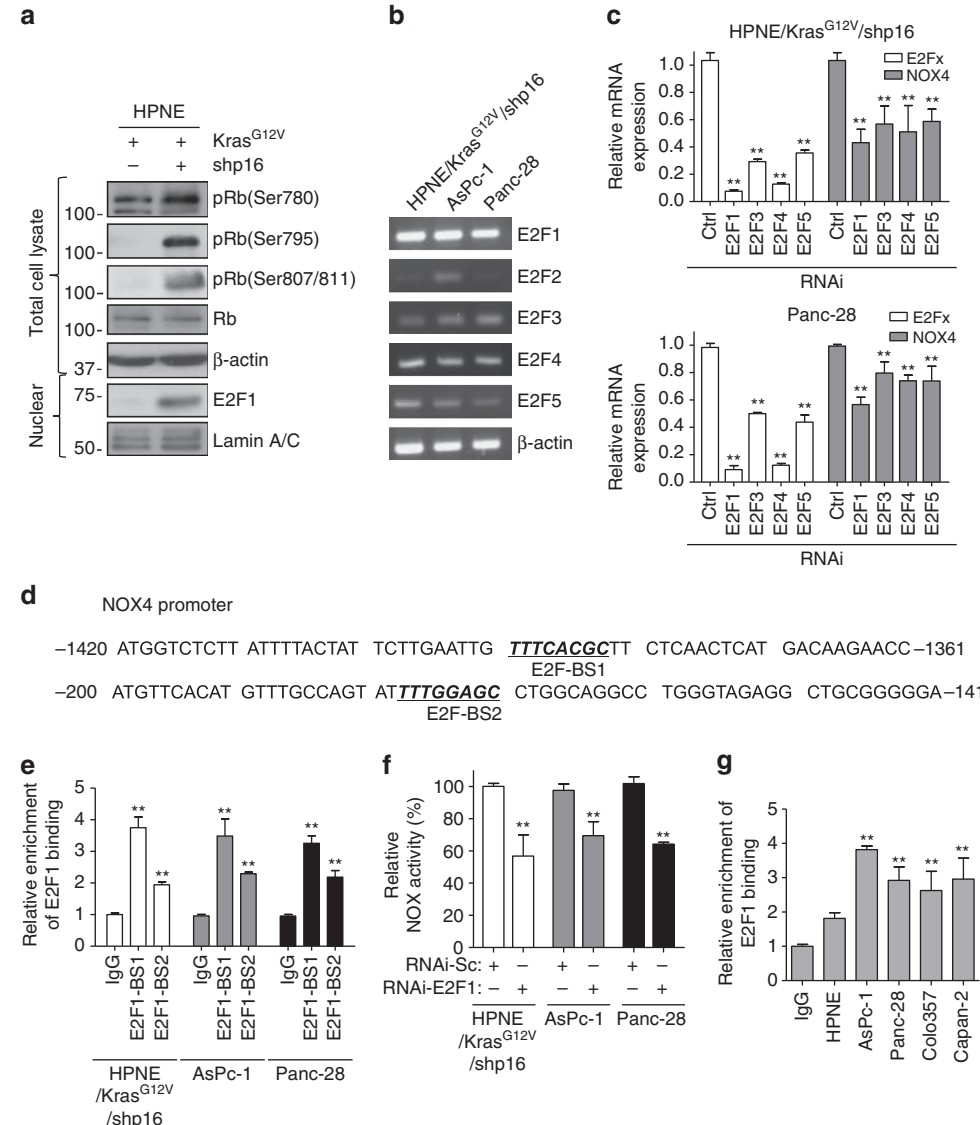

**Figure 5 | p16 upregulates NOX4 expression via the Rb-E2F pathway.** (**a**) The expression of p-Rb (Ser780), p-Rb (Ser795), p-Rb (Ser807/811), Rb and E2F1 in HPNE/Kras$^{G12V}$ and HPNE/Kras$^{G12V}$/shp16 cells was analysed by immunoblotting. β-actin and Lamin A/C was used as loading control. (**b**) The mRNA expression of E2F family members (E2F1, E2F2, E2F3, E2F4 and E2F5) in HPNE/Kras$^{G12V}$/shp16, AsPc-1 and Panc-28 cells was analysed by RT-PCR. (**c**) The expression of NOX4 was analysed by qPCR in HPNE/Kras$^{G12V}$/shp16 and Panc-28 cells transfected with siRNAs targeting different E2F family members. (**d**) The sequences of human NOX4 promoter regions are presented with E2F binding sites (BS-1 and BS-2) indicated. (**e**) The activities of E2F1 binding sites BS-1 and BS-2 in the NOX4 promoter were analysed in indicated cells by ChIP and qPCR assay. (**f**) NOX activity was determined in HPNE/Kras$^{G12V}$/shp16, AsPc-1 and Panc-28 cells transfected with E2F1 siRNA. (**g**) The activities of E2F1 binding sites BS-1 in the NOX4 promoter were analysed in indicated cells by ChIP and qPCR assay. IgG was used as a negative control. Data in **c,e,f,g** are presented as mean ± s.d. ($n = 3$). **$P < 0.01$ for indicated comparison (Student unpaired $t$-test).

recapitulate the key events in pathogenesis of human PDAC[6,31,32]. However, the biochemical mechanisms underlying this function have remained unclear. As reported here, we identified the mechanisms through which loss of p16 expression cooperates with oncogenic Kras in metabolic reprogramming from oxidative phosphorylation to glycolysis during PDAC development. Although it has been shown that inactivation of p16 and activation of Kras result in loss of a cell cycle checkpoint and stimulation of cell proliferation[33,34], our results have unequivocally demonstrated that loss of p16 and oncogenic activation of Kras induce NOX4 activity for metabolic reprogramming to maintain glycolysis and sustain cell growth, thus providing a novel mechanistic explanation for the requirement of inactivation of p16 and activation of Kras in PDAC development.

Seventy years ago, Warburg observed that tumours produce excess lactate in normoxic conditions and he interpreted this phenomenon as mitochondrial dysfunction[13,35]. As proposed by Vander Heiden et al. that cancer cells with a mitochondrial respiration defect, such as those caused by oncogenic Kras, require a high rate of glycolysis to produce sufficient ATP that promotes tumour progression[25,26], and incorporate nucleotides, amino acid and lipids required for generating a new daughter cell. NAD$^+$ is a crucial cofactor for the redox reactions in the metabolic pathways of cancer cells with elevated aerobic glycolysis[36]. The accumulation of lactate in tumours, which

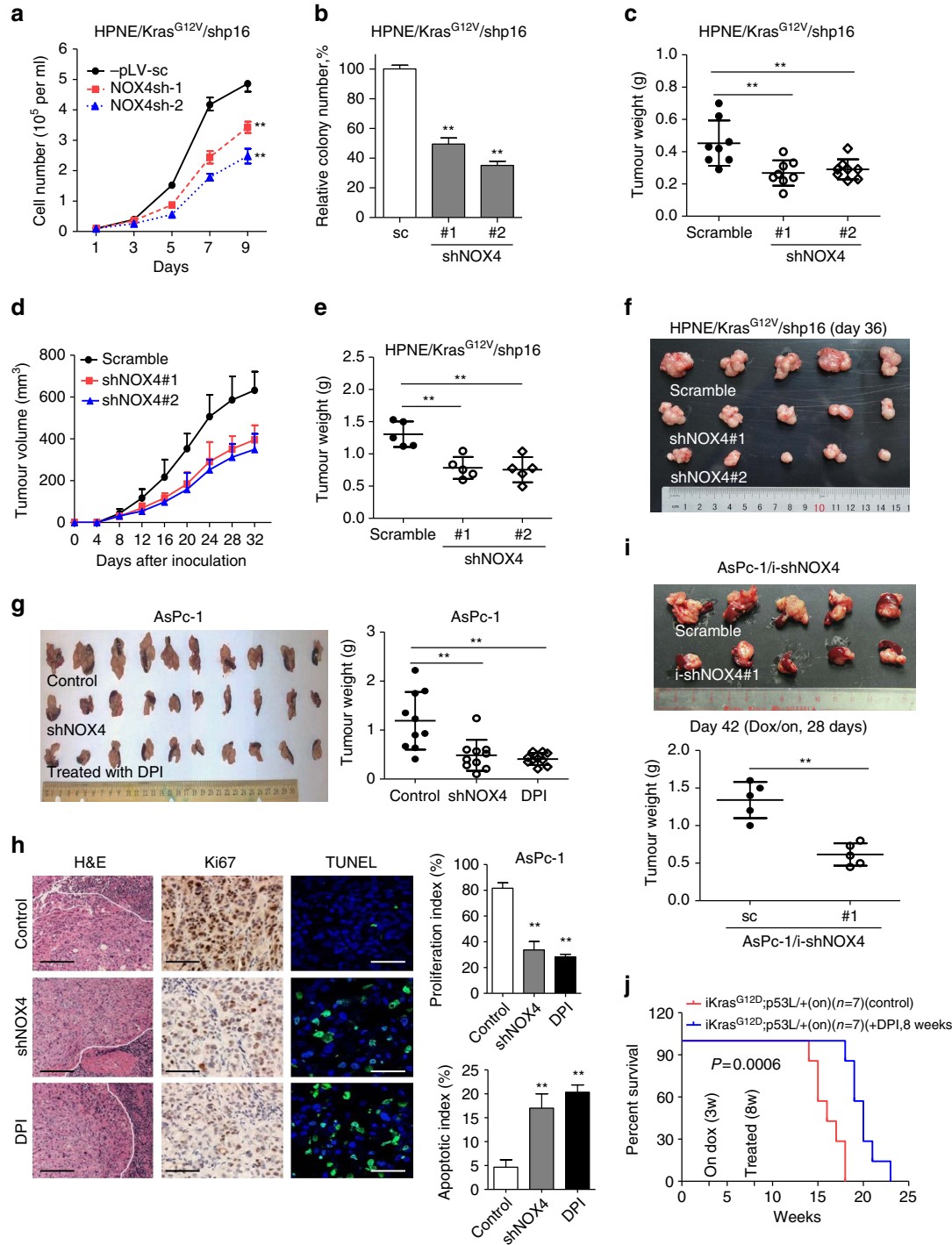

**Figure 6 | Suppression of NOX4 by shRNA or DPI inhibits PDAC growth *in vitro* and *in vivo*.** (**a,b**) Growth curves and clonogenic growth were measured in NOX4-silenced HPNE/Kras$^{G12V}$/shp16 cells. (**c**) Weights of PDAC tumours removed on day 60 from mice ($N=8$) injected orthotopically with NOX4-silenced and control HPNE/Kras$^{G12V}$/shp16 cells. (**d**) The nude mice were inoculated subcutaneously with indicated cells ($N=5$). The tumour sizes were measured throughout the experiment to evaluate NOX4 knockdown effects. (**e**) Tumour weight derived from indicated group was measured. (**f**) Photograph and comparison of excised tumour size. (**g**) Sizes and weights of PDAC tumours ($N=10$) removed on day 42 from mice injected orthotopically with NOX4-silenced AsPc-1 cells or control AsPc-1 cells treated with NOX inhibitor DPI (1.5 mg kg$^{-1}$ per mouse, i.p., 5 days per week). (**h**) Paraffin-embedded tumour sections were stained with H&E (P: PDAC; S: Spleen) or anti-Ki67 antibody (Scale bars, 100 μm); apoptotic cells were visualized by TUNEL staining (green) and counterstained with DAPI (blue) (Scale bars, 10 μm). Quantification of proliferation index and apoptotic index in PDAC tumours was shown. (**i**) Sizes and weights of tumour tissues ($N=5$) removed on day 42 from mice injected orthotopically with AsPc-1/i-shNOX4 cells. On: mice were fed with doxy-containing water from 2 weeks after inoculation, and continued for 4 weeks. **$P<0.01$ for indicated comparison (Student unpaired *t*-test). (**j**) Kaplan–Meier overall survival analysis for mice treated with DPI as indicated (Kaplan–Meier analysis with the log-rank test). On: mice were fed with doxy-containing water from 3 weeks of age. DPI: mice were treated with DPI (1.5 mg kg$^{-1}$ per mouse, i.p., 5 days per week) from 8 weeks of age. Data in **a,b** are presented as mean ± s.d. ($n=3$). Data in **c–e,g,i** are representative of two independent experiments and presented as mean ± s.d. **$P<0.01$ for indicated comparison (one-way analysis of variance (ANOVA) with the Newman Keul's multiple comparison test).

generate from higher glycolytic activity, implies an increase of NADH relative to $NAD^+$ and $NAD^+$ is insufficient in cancer cells. Also, lactate acts as strong inhibitors of PFK1 which can be overridden by the allosteric activator, fructose-2,6-bisphosphate $(F2,6BP)^{27}$. Our recent study showed that PDAC cells rely on the $NAD^+$ salvage pathway and reducing intracellular $NAD^+$ pools significantly affects glycolytic activity, cellular energy production and tumorigenesis in PDAC cells[37]. Here, we found that NOX4 activation by oncogenic Kras and inactivation of p16 enables cells to generate additional $NAD^+$ to support the glycolytic reaction catalysed by GAPDH and thus the highly active glycolysis required by PDAC cells. Our study showed that the $NAD^+$ generated by NOX4 was crucial to fuel glycolysis in PDAC cells. Suppression of NOX4 by siRNA in PDAC cells decreased NADH oxidation and cellular $NAD^+$ levels, glycolysis and cellular ATP production, and ultimately caused the interruption of cancer growth. The $NAD^+$ allows glycolysis to persist and the upregulation of NOX4 accelerates $NAD^+$ production. This role of NOX4 in supporting active glycolysis in cancer cell metabolism was previously unrecognized.

NOX4 is unique among the catalytic NOX subunits in requiring the membrane subunit p22$^{phox}$ for ROS-producing activity[17]. In cardiac fibroblasts and lung and pulmonary artery smooth muscle cells, transforming growth factor beta induces increased expression of NOX4 (ref. 15). Insulin-like growth factor I has been reported to activate NOX4 through transcriptional upregulation of p22$^{phox}$ via an Akt-dependent pathway in pancreatic cancer cells[19]. In those reports, however, the regulation mechanisms of NOX4 and p22$^{phox}$ expression is unexplained. The molecular mechanisms by which oncogenic Kras and inactivated p16 upregulate the expression of NOX4 and p22$^{phox}$ in PDAC cells have not been reported until now. With the present study, we unequivocally demonstrate that NOX4 activity is induced both through transcriptional upregulation of p22$^{phox}$ expression via oncogenic Kras-activated NF-κB, and though increasing NOX4 overexpression by loss of p16-regulated Rb-E2F pathway. Our findings not only reveal the novel function of NOX4 in metabolic reprogramming, but also provide a mechanistic explanation for why Kras activation and p16 inactivation are required for PDAC development.

Genetically suppression of NOX4 expression or pharmacologic targeting NOX4 using DPI leads to metabolic disruption and decreases pancreatic cancer growth *in vitro* and *in vivo*. It's worth mentioning that DPI is non-specific inhibitor for NOX4. It is a chemical inhibitor of flavoprotein-containing enzymes, including NOX oxidases[17]. Our finding also revealed that the tumour burden mice may develop resistance to DPI treatment after long time administration. Given that inhibition of NOX4 has proven potential for PDAC treatment and DPI just has moderate effects, the development of potent and specific drugs that target NOX4 deserves further exploration.

In summary, our findings illuminate that NOX4, which is regulated by oncogenic Kras and inactivation of p16 in PDAC, may play a role in promoting glycolysis through NOX4-dependent replenishment of $NAD^+$ levels, such that glycolysis continues despite PFK1 and LDHA inhibition by elevated lactate. As illustrated in Fig. 7, our findings reveal the novel function of NOX4 in reprogramming aerobic glycolysis initiated by activated Kras and inactivated p16 in PDAC, indicating its potential as a therapeutic target for PDAC and other cancers.

## Methods

**Cell lines and cell culture.** The human pancreatic cancer cell lines (AsPc-1, BxPc-3, Colo357, Capan-1, Capan-2, Panc-1, Panc-28) were purchased from the American Type Culture Collection (Manassas, VA, USA) and cultured under conditions specified by the supplier. PATC-43, PATC-50, PATC-53 and PATC-66, which were established from patient-derived xenografts (PDX), were provided by Dr Jason B. Fleming (MD Anderson Cancer Center, USA). The mouse PDAC cell line expressing inducible Kras$^{G12V}$ (mPDAC/iKras) was obtained from the laboratories of Drs Ying and DePinho[7]. The immortalized/nontumorigenic HPDE and HPNE cells were described elsewhere[4,38]. Other cell lines including Panc-28/IκBαM, AsPc-1/IκBαM (both, IκBα mutant), AsPc-1/iTak1sh, HPNE/Kras, HPNE/Kras$^{G12V}$/shp16, HPDE/Kras and HPDE/Kras$^{G12V}$/shp16 were established in Dr Chiao's laboratory and were cultured as described previously[4,5,39]. All cells were negatively tested for mycoplasma contamination either by the vendor or in house. All cell lines were authenticated by short tandem repeat (STR) fingerprinting before use at the Characterized Cell Line Core of MD Anderson Cancer Center.

**RNA isolation and qPCR with reverse transcription.** Total RNA was isoltated with TRIzol regent (# 15596-08, Life Technologies, Carlsbad, USA) and then reverse transcribed with an iScript cDNA Synthesis Kit (Bio-Rad). The resulting complementary DNA was analysed by qPCR performed with SYBR reagent using the IQ5 PCR system (Bio-Rad), and data were normalized to β-Actin. Specific primers were synthesized by Sigma-Aldrich (Louis, MO, USA) and the sequences were described in Supplementary Table 2. The qPCR primers used for CHIP assay were described as follows: p22$^{phox}$ promoter NF-κB/p65 Site 1: 5'-tccgccgtttt tctgtttcg-3', 5'-ggaaagcacagaatgcaggg-3'; NF-κB/p65 Site 2: 5'-ccttcacaccttgtcctgc t-3', 5'-gcatctgtagggtgcaggg-3'; NOX4 promoter E2F1 Site 1: 5'-agtgctaacacgcac atgga-3', 5'-atagatgggggcaggaggtt-3'; E2F1 Site 2: 5'-gcgagggtccccacttttag-3', 5'-cctttgtctaggggcgagc-3'.

**Immunoblotting analysis.** Immunoblotting analyses were performed with precast gradient gels (Bio-Rad) using standard methods. Briefly, cells were lysed in radioimmunoprecipitation assay (RIPA) buffer and normalized using a BCA protein assay kit (Thermo Scientific). Proteins were separated by sodiumdodecyl sulphate-polyacrylamide gel electrophoresis (SDS–PAGE) and blotted onto a PVDF membrane (Millipore). Membranes were probed with the specific primary antibodies and then with peroxidase-conjugated secondary antibodies. The bands were visualized by enhanced chemiluminescence using Hyperfilm ECL. Uncropped images of immunoblots presented in the main paper are provided in Supplementary Fig. 7. The following antibodies were used: antibodies against NOX4 (1:2,000, ab133303), p16 (1:2,500, ab51243), p22$^{phox}$ (1:1,000, ab80896) (Abcam, Cambridge, USA); Tak1 (1:1,000, #5206), Phospho(p)-Tak1 (1:1,000, #4508), NF-κB/p65 (1:1,000, #8242), Phospho(p)-NF-κB/p65 (1:1,000, #3039), Lamin A/C (1:1,000, #4777), p-Rb (Ser780, 1:1,000, #9307), p-Rb (Ser795, 1:2,000, #9301), p-Rb (Ser807/811, 1:1,000, #8516), Rb (1:1,000, #9309) and E2F1 (1:1,000, #3742) (Cell Signaling Technology, Beverly, MA, USA); Kras (1:500,sc-521), IκB-α (1:200, #SC-371), (Santa Cruz Biotechnology, Santa Cruz, USA) and β-actin (1:20,000, #5316) (Sigma-Aldrich, St Louis, MO).

**Determination of NOX activity and NAD(P)H/NAD(P)$^+$ levels.** NOX activity was measured by a lucigenin-derived chemiluminescence assay[23]. Briefly, 5 μg of homogenized protein was incubated with its substrate 100 μM NADPH or NADH in a phosphate buffer (50 mM, pH 7) containing 150 mM NaCl and 1 mM EGTA for 15 min, followed by addition of 5 μM lucigenin and incubation for 15 min in the dark. The chemiluminescent signal was measured by a Turner 20/20 luminometer (Turner Designs). The intracellular levels of NADH/NAD$^+$ and NADPH/NADP$^+$ was measured with a NADH/NAD$^+$-Glo Assay and NADPH/NADP$^+$-Glo Assay kits (Promega Corporation, Madison, WI, USA), respectively, according to the manufacturer's instructions.

**Determination of intracellular ROS and GSH/GSSG levels.** Flow cytometry determination of cellular $O_2^-$ was performed with a BD Biosciences FACSCanto flow cytometer (BD Biosciences, San Jose, CA, USA), and the data were analysed by the Flow-Jo software. Briefly, cells were seeded in 6-well plates overnight, then incubated with 200 ng ml$^{-1}$ HEt at 37 °C for 60 min, protected from light. After incubation, cells were harvested with trypsin, washed twice with PBS and evaluated by flow cytometry. The intracellular levels of GSH/GSSG was measured with a GSH/GSSG-Glo Assay kit (Promega, WI, USA) according to the manufacturer's instructions.

**ChIP assay.** Cells were grown to 80% confluence, and crosslinking was performed with 1% formaldehyde for 10 min. ChIP assays were performed using a Chromatin Immunoprecipitation Assay Kit (Millipore, Bedford, MA, USA) according to the manufacturer's instructions. After immunoprecipitation with an antibody against NF-κB/p65 or E2F1 (Cell Signaling) or normal rabbit IgG, protein/DNA crosslinks were reversed. DNA was then purified to remove the chromatin proteins and analysed by qPCR to amplify the corresponding region in p22$^{phox}$ promoter containing the two NF-κB binding sites and NOX4 promoter containing the two E2F binding sites.

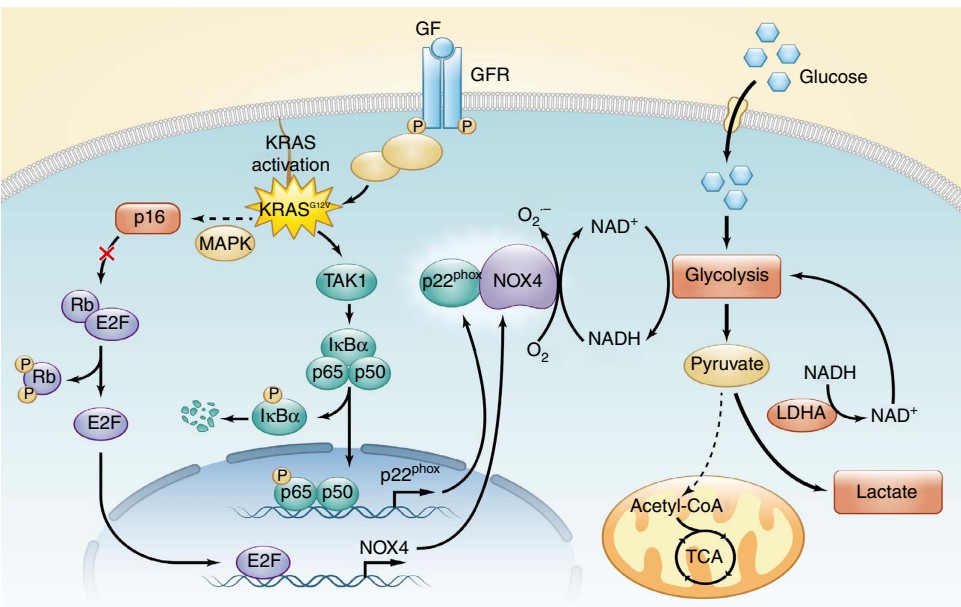

**Figure 7 | Mutant Kras and inactivated p16 orchestrate metabolic reprogramming in PDAC.** The proposed working model of the function and mechanisms of NOX4 in reprogramming aerobic glycolysis initiated by activated Kras and inactivated p16 in PDAC.

**Metabolic measurements.** For measuring oxygen consumption (OCR), cells were trypsinized and resuspended in 1 ml fresh culture medium pre-equilibrated with 21% oxygen at 37 °C; the cells were then transferred to the sealed respiration chamber of a Clark-type oxygen measuring system (Oxytherm, Hansatech Instruments) Cellular glucose uptake was determined by using the Amplex Red Glucose Assay Kit (Invitrogen) according to the manufacturer's instructions. Briefly, cells in exponential growth phase were washed and incubated in fresh medium. Medium was collected and diluted 1:400 in water. Glucose uptake was determined by the difference in glucose concentration of each sample compared with that of the control medium without cells. To measure lactate production, cells with 80% confluency were replenished with fresh medium. Aliquots of the medium were removed after 12 h for measurement of lactate with an Accutrend lactate analyzer (Roche). Cells were also counted for normalization of glucose uptake and lactate generation. Cellular ATP contents were measured by using the CellTiter-Glo Luminescent Cell Viability Assay kit (Promega) according to the manufacturer's recommendations.

**Isotope labelling and metabolite analysis.** Cells were seeded in 6-well plates overnight, and replaced with medium containing U-$^{13}C_6$ glucose. After 1 and 3 h, cells were washed with cold PBS once and quenched with 400 µl of cold methanol. Same volume of water containing 1 µg of norvaline was added, and cells were scraped into 2 ml tubes. Then, the chloroform (800 µl) was added. Solution was vortex at 4 °C for 30 min, and centrifuged at 7,300 r.p.m. for 10 min at 4 °C. The aqueous layer was collected for metabolite analysis[40,41]. Brifly, aqueous samples were dried through speedvac and dissolved in 30 µl of 2% methoxyamine hydrochloride in pyridine (Pierce), and sonicated for 10 mins. Afterwards, samples were kept in 37 °C for 2 h and kept for another 1 h at 55 °C after addition of 45 µl of MBTSTFA + 1% TBDMCS (Pierce). Samples were transferred into GCMS running vials (Thermo Fish Scientific). GC/MS analysis was performed using an Agilent 6890 GC equipped with a 30-m Rtx-5 capillary column for metabolites samples, connected to an Agilent 5975B MS. The abundance of pyruvate and lactate was calculated from the integrated signal of all potentially labelled ions for each metabolite fragment.

**RNA-mediated interference.** Cells were seeded in 6-well plates at $1 \times 10^5$ cells per well, respectively, and transfected using DharmaFECT transfection reagent (Dharmacon, Lafayette, USA) with 20 nM siRNA targeted against indicated genes or a scrambled RNA negative control. Cells were transfected for the specified time periods depending on the subsequent analyses. ON-TARGET siRNA targeting against the following genes were purchased from Dharmacon, including NOX4 (Target sequence: #1 5′-acuaugauaucuucuggua-3′; #2 5′-gaaauuaaucccaagcugua-3′); p22$^{-phox}$ (Target sequence: #1 5′-gaagaagggcuccaccaug-3′; #2 5′-guacaugac cgccguggug-3′); p16 (Target sequence: #1 5′-gaucaucagucaccgaagg-3′; #2 5′-aaacac cgcuucugccuuu-3′); NF-κB/p65 (Target sequence: #1 5′-gcccuaucccuuuacguca-3′; #2 5′-gagcaccaucaacuaugaugaguuu-3′); E2F1 (Target sequence: 5′-ucggagaacuu ucagaucu-3′); E2F3 (Target sequence: 5′-cuucaugaguguaguugauua-3′); E2F4 (Target

sequence: 5′-cgggagaccacgauuauau-3′); E2F5 (Target sequence: 5′-ggugcuggcugua auacua-3′). Rescue experiments were performed by co-expressing human NOX4 siRNA and siRNA resistant NOX4 (NOX4-R), which was synthesized by GeneCopoeia (Rockville, USA) in AsPc-1 and Panc-28 cells.

**Lentiviral transduction.** The human NOX4 gene was PCR-amplified from cDNA and DNA sequences encoding shRNAs targeting NOX4 were chosen to clone into the pLV lentiviral vector (Biosettia). Retroviral production and infection were performed according to the manufacturer's instructions (Biosettia)[5,37]. Briefly, lentiviruses were generated by transfecting lentiviral vectors together with packaging vector psPAX2 and envelope plasmid pMD2.G into 293T cells. After 48–72 h transfection of lentiviral vectors, viral culture supernatants from the 293T cells were collected and the cells were infected by lentiviruses containing the shRNA. Stable cell lines expressing NOX4 or NOX4 shRNAs were selected for 10 days with puromycin or sorted by red fluorescent protein 4 days after infection, respectively. The expression levels of NOX4 in transfected cells were determined by qPCR and immunoblotting. Oligo sequences for NOX4 shRNA were (#1) 5′-gctgtatattgatggtcct-3′ and (#2) 5′-gctgaagtatcaaactaat-3′.

**In vivo tumorigenesis study.** The experiments with mouse xenografts were carried out according to protocols approved by the Institutional Animal Care and Use Committee (IACUC) of MD Anderson Cancer Center. Male C57BL/6 mice aged 5 to 6 weeks were purchased from Charles River Laboratories International. When cell confluence reached 80%, cells were harvested and washed twice with PBS buffer. Cells were suspended in PBS buffer ($1 \times 10^6$ cells per 50 µl) with 10% Matrigel. For orthotopic implantation, each mouse was injected with $1 \times 10^6$ cells. The mice were killed and the tumour growth was analysed 8 weeks after injection. Moribund animals were killed as mandated by the IACUC protocol, and the tumour images and weights were recorded. To show the kinetics of tumour progression, xenograft tumours were generated by subcutaneous injection of cells into the flanks of nude mouse at $2 \times 10^6$ cells per injection site. Tumour volumes were measured and calculated as described previously[42]. Data of 5–10 animals per experimental group are indicated in figure legends. Randomization was conducted by using an internal computer software (Indigo). The transgenetic iKras; p53$^{L/+}$ mice generated from crossing iKras and conditional p53 knockout (p53$^L$) alleles[7], and maintained in pathogen-free conditions at MDACC Institute. For Doxtreatment, mice were fed with Dox water (doxy 2 g l$^{-1}$, sucrose 20 g l$^{-1}$).

**Tissue microarray and IHC analysis.** The pancreatic cancer tissue microarray was constructed and paraffin sections were obtained from paraffin blocks of primary PDAC and paired adjacent normal tissues of 131 pancreatic cancer patients. Tissue specimens were collected within 1 h after surgery under a protocol approved by the Institutional Review Board at MD Anderson Cancer Center, and written informed consent was obtained from all patients at the time of enrolment. Tissue sections were subjected to standard IHC analysis and photographed by a digital camera attached to the microscope. The sections were then incubated with a rabbit

polyclonal antibody against NOX4 (Abcam, 1:750 dilution). The staining results were evaluated by a gastrointestinal pathologist (H. Wang).

**Histopathological analysis.** Mouse pancreas tumours were fixed in 10% formalin, and embedded in paraffin. Sections were stained with hematoxylin and eosin (HE) according to standard procedures as previously described[5]. For IHC analysis of Ki67 and NOX4, standard procedures were carried out according to the manual (#PK-6101 and #PK-6102, Vector Laboratories). Briefly, the sections were de-waxed and rehydrated. After being retrieved antigen via unmasking solution (#H3300, Vector Laboratories) in a steamer, slides were blocked with biotin blocking solution (Avidin/Biotin blocking kit, SP2001, Vector Laboratories) and 1% BSA 30 min, respectively; and then incubated with an anti-Ki67 (#RM-9106-S0, Thermo Scientific) and NOX4 (ab133303, Abcam) antibodies at 4 °C overnight. Next day, all these slides were processed by ABC method (Vector Laboratories) as described in the manufacturer's instruction. Slides were subsequently incubated with biotinylated secondary antibody and ABC reagent at room temperature for 30 min, respectively. 3,3′-diaminobenzidine (DAB, Vector Laboratories) was then incubated as the final chromogen, the staining time were 2 to 5 min to see brown colour under the microscope. Finally slides were counterstained with hematoxylin, and then serially dehydrated, and sealed with permount medium after air drying. All samples were processed under the same conditions.

**Statistical analysis.** For error bars in all experiments, s.d. was calculated from three independent experiments and values represent mean ± s.d. The variance between the groups that are being statistically compared is similar. For comparison of the statistical differences among more than two groups, one-way ANOVA and the Newman Keul's multiple comparison test was used. All other differences were evaluated by the Student unpaired $t$-test. Survival curves were generated using the Kaplan–Meier method and assessed using the log-rank test. Sample size was chosen based on the need to have sufficient statistical power. Differences reached statistical significance with $P < 0.05$ (*) and $P < 0.01$ (**). Statistical computations were performed using Prism software (Graph Pad, La Jolla, CA, USA).

**Data availability.** The gene expression data referenced in Fig. 2g are available in multiple cancer microarray data sets from Oncomine (https://www.oncomine.com//). 'NOX4' or 'p22phox' was used as keyword in the Oncomine query, 'Cancer versus Normal Analysis' was used as the primary filter and 'Pancreatic Cancer' was choosed as the cancer type. The cDNA microarray related to Fig. 1b were deposited to Gene Expression Omnibus (GEO), the record number is GSE89422; Link: http://www.ncbi.nlm.nih.gov/geo/query/acc.cgi?acc=GSE89422. All the other data supporting the findings of this study are available within the article and its Supplementary Information Files or from the corresponding authors upon reasonable request.

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

## Acknowledgements

We thank Dr Wei Zhang of the Genomic Core facility and Dr Jun Yao of the department of Molecular and Cellular Oncology, MD Anderson Cancer Center, for microarray analysis. This work was supported in part by the National Institutes of Health through The University of Texas MD Anderson Cancer Center Support Grant (CA016672) and

through grants CA097159, CA142674 and CA109405 (to P.J.C.); by grants from the Skip Viragh Family Foundation (to P.J.C.). This work was also supported by grants from the China Postdoctoral Science Foundation (Nos. 2015M570746 and 2016T90818) and the National Natural Science Foundation of China (No. 81430060).

## Author contributions

H.-Q.J., P.H. and P.J.C. conceived the project, designed the experiments and wrote the manuscript. H.-Q.J., H.Y., T.T., J.L., J.F., M.W., G.C. and Z.Z. performed the experiments. H.W. provided the TMA of PDAC samples and analysed the IHC data. D.N. provided the method and expertise for isotope labelling and metabolite analysis and data interpretation; M.-C.H., R.A.D. and R.-H.X. provided very important comments and suggestions.

## Additional information

**Competing financial interests:** The authors declare no competing financial interests.

