## [Peer Review File · Nature Communications]

Reviewers' comments:

Reviewer #1 : Expert in cancer metabolism
(Remarks to the Author):

In this manuscript the authors report on the role of KRas activation and the role of NOX4 in malignant transformation of HPNE cells. They report on the role of the increased NOX4 activity that appears linked to overcoming metabolic checkpoints, resulting in formation of pancreatic ductal carcinomas. This is an interesting paper that needs some additional work, as indicated below.

The authors show that Kras results in increased p16, and that this protein needs to be silenced for the HPNE cells to undergo malignant transformation. If this is so, how would Kras result in malignant transformation in patients? Do they have silenced p16? Or, is it known to be silenced in models of malignant transformation of normal cells? What is the pathophysiological context of the current research? It would be interesting (and should be checked) whether p16 is 'silenced' in pre-malignant lesions. More realistic models should be used, the HPNE cells with silenced p16, albeit providing interesting results, are an artificial system. Although there are references to the link of p16 suppression and malignant transformation, the authors should show this in the context of their research.

On a more general level, the authors had shown before that KRas causes a switch in cells from respiration to glycolysis. They should refer here to work by others, since this topic has been covered extensively in the past, and is rather controversial (e.g. Weinberg et al PNAS 2010, 1078, 8788).

There is no attempt in the report to characterize respiration of the manipulated cells. That is, HPNE KRas-transfected and p16-silenced cells. This should be included.

The results in Fig. 6 concerning tumour formation are insufficient. There is only the endpoint results, tumour weight, documenting the differences. The authors should show kinetics of tumour progression. They should also analyse the tumours derived from control and shNOX4 cells for glycolysis and respiration.

Reviewer #2: Expert in PDAC and Ras signalling
(Remarks to the Author):

The current manuscript by Dr Chiao and colleagues addresses the role of Nox4 in pancreatic cancer. Using HPNE and HPDE cells, they show that Nox4 expression is up-regulated upon activation of oncogenic Kras and concomitant loss of the tumor suppressor p16. Further, they show that the catalytic subunit of Nox4, p22PHOX, is expressed downstream of oncogenic Kras through activation of NFκB. They further show that Nox4 overexpression is detected in human pancreatic cancer samples, and that it mediates metabolic reprogramming that is characteristic of pancreatic cancer. Finally, the authors provide evidence that Nox4 inhibition is cytostatic in pancreatic cancer cell lines, and that the Nox4 inhibitor DPI increases survival in the iKras mouse model of pancreatic cancer. The manuscript investigates a timely topic, namely metabolic reprogramming in pancreatic cancer and its potential therapeutic implications. Some additional analysis could strengthen the underlying message.

1) The authors propose that Nox4 overexpression requires concomitant loss of p16 and activation of oncogenic Kras. They should determine whether this holds true in the panel of human cell lines and tissue samples that are included in Figure 2A, B. In addition, they should investigate the expression of Nox4 in iKras mouse tumors (Figure 6), which have Kras expression combined with inactivation of p53, rather than p16.

2) In Figure 4, the authors should determine whether modulation of Kras and its downstream effectors alters the expression of Nox4, in addition to p22PHOX.

3) The data indicating reduced tumor growth upon inactivation of Nox4 by shRNA could be

strengthened by the use of an inducible shRNA, which would allow to determine whether the effect of Nox4 inactivation is due to a defect at the implantation stage or in the maintenance/growth of established tumors.

4) The increased survival in iKras mice (Figure 6G) is impressive, but would benefit from further analysis. It appears that the survival curve is pushed to the right, yet once the mice start dying the slope is similar as to the control. At the very least histological analysis of the tumor should be performed to determine whether resistance mechanisms have been established. An easy readout would be to determine whether the proliferation index has rebounded. Another possibility is that the tumor grows more slowly, thus reach the critical point later. Understanding a potential mechanism of resistance would be important to assess any therapeutic outcome.

5) On a different note, related to the iKras experiment, the authors should comment on the specificity of DPI as a Nox4 inhibitor.

Point-by-point replies to the reviewers' comments.

Reviewers' comments:

Reviewer #1: Expert in cancer metabolism (Remarks to the Author):

In this manuscript the authors report on the role of KRas activation and the role of NOX4 in malignant transformation of HPNE cells. They report on the role of the increased NOX4 activity that appears linked to overcoming metabolic checkpoints, resulting in formation of pancreatic ductal carcinomas. This is an interesting paper that needs some additional work, as indicated below.

Response: Thank you very much for reviewing our manuscript. We are very glad to know that the reviewers thought our findings are interesting and significant. We appreciate the reviewers' questions and comments, which have been replied with a point by point response. All changes are identified by page and paragraph locations, and noted by highlight or strikethrough in the text. Our detailed responses are as follows.

1) *The authors show that Kras results in increased p16, and that this protein needs to be silenced for the HPNE cells to undergo malignant transformation. If this is so, how would Kras result in malignant transformation in patients? Do they have silenced p16? Or, is it known to be silenced in models of malignant transformation of normal cells? What is the pathophysiological context of the current research? It would be interesting (and should be checked) whether p16 is 'silenced' in pre-malignant lesions.*

Response: That is a good point and well taken. We apologize for not providing a clear explanation and evidence. As illustrated in the following progression model of PDAC, the mutational activation of Kras is an early event in PDAC development and

has been detected in approximately 95% PDAC (Bardeesy N, DePinho RA. Nat Rev Cancer 2, 897-909, 2002), and mutational inactivation of p16^{INK4a} can be identified in approximately 80%–90% of PDAC (Schutte M., Cancer Res. 57, 3126–3130, 1997). Our previous experimental results have demonstrated that silencing p16^{INK4a} expression in HPNE/Kras^{G12V} cells resulted in tumorigenic transformation and PDAC development in cell culture model and orthotopic mouse model (Chang Z, et al. PLoS One 9, e101452, 2014). Also, genetically engineered mouse models (GEMM) with mutant Kras have unequivocally demonstrated that additional inactivation of p16^{INK4a} dramatically accelerated the progression of Kras initiated PDAC (Aguirre AJ, et al. Genes Dev 17, 3112-3126, 2003). These evidences suggest that p16^{INK4a} alterations plays an essential role in development of pancreatic cancer through its interaction with various cellular signaling pathways.

Progression model of pancreatic ductal adenocarcinoma from normal epithelium to invasively growing tumor (Ottendorf et al, Pathology Research International. 2011).

As suggested, we also checked whether p16^{INK4a} is 'silenced' in PDAC development. The loss of p16/INK4a expression can be unequivocally demonstrated in PanIN or PDAC tissues (21 patient samples), but it was detectable in normal duct cells analyzed by IHC staining (Supplementary Fig. 1a). Also, the mRNA level of p16 was only detected in three cell lines (AsPc-1, Colo357 and Capan-2) and one out of 21 PDAC patient samples by qPCR

assays, consistent with the previous reports (Supplementary Fig. 1b). These results are shown in Supplementary Figure 1 as below and described in our revised manuscript (Page 5, paragraph 2).

Supplementary Figure 1 (a) Representative IHC staining shows the loss of p16 expression in PanIN (pancreatic intraepithelial neoplasia) and PDAC tissues (21 patient samples). (b) The expression of p16 in PDAC cell lines and tissues was analyzed by qPCR assay. Data in B are presented as mean \pm SD (n=3). ** $P < 0.01$.

2) *More realistic models should be used, the HPNE cells with silenced p16, albeit providing interesting results, are an artificial system. Although there are references to the link of p16 suppression and malignant transformation, the authors should show this in the context of their research.*

Response: We thank the reviewer for the suggestion. So far, HPNE is the one of the two immortalized/nontumorigenic pancreatic epithelial cell lines (Lee et al., BBRC, 2002; Qian et al Cancer Res. 2005). One striking difference between our previous and current studies is that the previous studies depended on the expression of DNA tumor viral oncogenes, such as the SV40 early region encoding both the large T and small t antigens and HPV E6E7 genes, to induce malignant transformation

[Campbell et al., Cancer Res. 2007; Hahn et al. Nature, 1999; Elenbaas et al Genes Dev 2001; Hahn et al, MCB 2002]. These viral genes are not associated with most human cancers. Moreover, the cellular target and function of these viral genes in malignant transformation remain poorly defined, which makes the study of the mechanisms of transformation more complicated. Our model uses the signature alterations in PDAC, which are the three most common gene alterations seen in PDAC: hTERT, mutant K-ras, and silencing of p16 to acquire immortality, sustained proliferative signaling, evasion of growth suppressors, disruption of the senescence checkpoint and ultimately, tumorigenesis, thus providing a useful cell culture model to study tumorigenic mechanism (Chang et al., Clinical Cancer Res., 2014). Our results suggest that constitutively activated Kras^{G12V} is strongly associated with loss of p16 expression, which required for tumorigenic transformation as demonstrated in HPNE cells and genetically engineered mouse models (GEMM) mouse model.

To determine whether the expression of NOX4 was increased in GEMM from *Pdx1-Cre; Kras^{G12D}; p16^{F/F}* mice, immunohistochemical analyses were performed. Levels of NOX4 expression were substantially higher in these tumors than in histologically normal pancreata from control mice (Figure 2e). Thus, the expression of NOX4 was also increased in PDAC from the PDAC genetically engineered mouse model. These results are shown in our revised Figure 2e and described in our

revised manuscript (Page 8, paragraph 2).

Figure 2 (e) Representative IHC staining with H&E or anti-NOX4 antibody in sections of formalin-fixed tissue from Pdx1-Cre; Kras^{LSL-G12D}; Ink4a^{F/F} mice wild-type mice.

3) *On a more general level, the authors had shown before that KRas causes a switch in cells from respiration to glycolysis. They should refer here to work by others, since this topic has been covered extensively in the past, and is rather controversial (e.g. Weinberg et al PNAS 2010, 1078, 8788).*

Response: We apologize for citing the references one-sidedly. We have added the reviewer recommended literature (Weinberg et al PNAS 2010, 1078, 8788) in our revised manuscript (Page 3, paragraph 1).

“Recent studies by our group and others showed that Kras activation led to suppression of mitochondrial respiratory activity and rendered the cell more dependent on glycolysis [1, 2]. Conversely, others reported that mitochondrial ROS generation is essential for Kras-induced cell proliferation and Kras-mediated tumorigenicity [3]. Dysfunctional mitochondria and increased aerobic glycolysis are two important biochemical characteristics observed frequently in cancer cells [1, 4, 5]. A metabolic switch from oxidative phosphorylation in the mitochondria to glycolysis in the cytosol in cancer cells has been well known as “Warburg effect” for decades [6, 7].”

4) *There is no attempt in the report to characterize respiration of the manipulated cells. That is, HPNE KRas-transfected and p16-silenced cells. This should be included.*

Response: Following the reviewer's suggestion, the respiration of the HPNE, HPNE/Kras^{G12V}, and HPNE/Kras^{G12V}/shp16 cells were detected though measuring the oxygen consumption rate. As expected, the mitochondrial respiratory chain activity was reduced in HPNE/Kras^{G12V} and HPNE/Kras/shp16 cells as evinced by substantial decreases in oxygen consumption rate (OCR). These results are shown in our revised Figure 3a as below and described in our revised manuscript (Page 9, paragraph 2; Page 18, paragraph 3).

Figure 3 (a) Oxygen consumption rate was determined in HPNE, HPNE/Kras, HPNE/Kras/shp16 cells.

To strengthen our experimental evidence, we performed metabolite isotope tracing experiments with 13 carbon labeled glucose (U-¹³C₆ Glu). We found that overexpression of NOX4 and Kras^{G12V}/shp16 in HPNE cells increased pyruvate and lactate level, thereby confirming the increased glucose to lactate conversion or glycolysis. Although the data from the described tracing experiments are not requested directly by reviewers, the results strengthen our finding and conclusion. So, these results are added in our revised Figure 3G-3H described in our revised manuscript (Page 10, paragraph 2), and the detail method are shown in our revised supplementary information.

Figure 3 (G) The expression level of NOX4 was analyzed by qPCR and immunoblotting in NOX4-overexpressed HPNE cells. (H) The pyruvate and lactate levels were measured in HPNE/NOX4 or HPNE/Kras^{G12V}/shp16 cells compared with parental HPNE cells using metabolite isotope tracing experiments with 13 carbon labeled glucose (U-¹³C₆ Glu). M3: three ¹³C-labeled carbons. Data in g, h are presented as mean ± SD (n=3). **P < 0.01.

5) *The results in Fig. 6 concerning tumour formation are insufficient. There is only the endpoint results, tumour weight, documenting the differences. The authors should show kinetics of tumour progression.*

Response: We thank the reviewer for the suggestion. To show the kinetics of tumor progression, xenograft tumors were generated by subcutaneous injection of cells (HPNE/Kras^{G12V}/shp16-shNOX4) into the flanks of each 4~6 week old Balb/C athymic nude mouse (nu/nu) at 2.0×10⁶ cells per injection site (N=5). Tumor size was measured by a slide caliper every 4 days and tumor volume was determined by the formula V=length×width²/2. Similarly, we observed that the tumors formed by HPNE/Kras^{G12V}/shp16-shNOX4 (#1 and #2) cells grew at a much slower rate than control HPNE/Kras^{G12V}/shp16-scramble tumors. Collectively, these results indicate that NOX4 plays a significant role in the tumorigenicity of PDAC cells *in vivo*. These results are shown in our revised Figure 6d-6f as below and described in our revised manuscript (Page 13, paragraph 2).

Figure 6 (D) The nude mice were inoculated subcutaneously with indicated cells (2.0×10^6). The tumor sizes were measured throughout the experiment to evaluate NOX4 knockdown effect. (E) Tumor weight derived from indicated group was measured. (F) Photograph and comparison of excised tumor size.

6) They should also analyze the tumors derived from control and shNOX4 cells for glycolysis and respiration.

Response: We thank the reviewer for the suggestion. To analyze the tumors derived from control and shNOX4 cells for glycolysis and respiration, we established and utilized the early passage of primary culture cells derived from control and shNOX4 subcutaneous xenograft mice. As expected, we observed that knockdown of NOX4 significantly decreased the glucose uptake and lactate production levels, but increased the oxygen consumption rate in HPNE/Kras^{G12V}/shp16 cells. These results are shown in Supplementary Figure 5 B as below and described in our revised manuscript (Page 14, paragraph 1).

B
Supplementary Figure 5 (B) Glucose uptake and lactate production, and oxygen consumption rate were measured in primary HPNE/Kras^{G12V}/shp16-scramble or HPNE/Kras^{G12V}/shp16-NOX4sh (#1 and #2 cells) derived from subcutaneous xenograft mice. Data are presented as mean \pm SD (n=3). ** $P < 0.01$.

Reviewer #2: Expert in PDAC and Ras signalling (Remarks to the Author):

The current manuscript by Dr Chiao and colleagues addresses the role of Nox4 in pancreatic cancer. Using HPNE and HPDE cells, they show that Nox4 expression is up-regulated upon activation of oncogenic Kras and concomitant loss of the tumor suppressor p16. Further, they show that the catalytic subunit of Nox4, p22PHOX, is expressed downstream of oncogenic Kras through activation of NFkappaB. They further show that Nox4 overexpression is detected in human pancreatic cancer samples, and that it mediates metabolic reprogramming that is characteristic of pancreatic cancer. Finally, the authors provide evidence that Nox4 inhibition is cytostatic in pancreatic cancer cell lines, and that the Nox4 inhibitor DPI increases

survival in the iKras mouse model of pancreatic cancer. The manuscript investigates a timely topic, namely metabolic reprogramming in pancreatic cancer and its potential therapeutic implications. Some additional analysis could strengthen the underlying message.

Response: Thank you very much for your review of our manuscript. Thank you for your comments and the following suggestion to improve our study. All changes are identified by page and paragraph, and noted by strikethrough or highlight in the text. Below are our detailed point by point responses.

- 1) *The authors propose that Nox4 overexpression requires concomitant loss of p16 and activation of oncogenic Kras. They should determine whether this holds true in the panel of human cell lines and tissue samples that are included in Figure 2A, B. In addition, they should investigate the expression of Nox4 in iKras mouse tumors (Figure 6), which have Kras expression combined with inactivation of p53, rather than p16.*

S1b

S2c

Supplementary Figure 1 (b) The expression of p16 in PDAC cell lines and tissues was analyzed by qPCR assay. Data in B are presented as mean \pm SD (n=3). **Supplementary Figure 2 (b)** Representative IHC staining with H&E or anti-NOX4 antibodies in sections of formalin-fixed PDAC from transgenic iKras; p53^{L/+} mice.

Response: That is a good point and well taken. As suggested, we firstly detected the expression profile of p16 in 12 PDAC cell lines and 21 human PDAC tissues by qPCR assay. We found that the mRNA level of p16 was only detected in one of 21 patient PDAC samples, and only expressed in AsPc-1, Colo357 and Capan-2 cell lines (Supplementary Fig. 1b). These results demonstrated that p16 is mutational inactivated in most PDAC. However, even p16 is high expressed in the Colo357 and Capan-2 cells, the NOX4 expression level was still high. Considering that gene transcription is a complicated process, it is likely that other transcription factors and co-activators are involved in the regulation of NOX4. P53 are reported to regulate

NOX4 expression in lung and breast cancer (Br J Cancer. 2014 May 13;110 (10):2569-82). As known, p53-regulated p21 expression is decreased when p53 is knocked out. As the result, CDK4 is activated and phosphorylates Rb, thus, leading to activation of E2F. Indeed, IHC staining indicated that the expression of NOX4 was substantially higher in these PDAC from iKras; p53^{L/+} mouse model than in histologically normal pancreata from control mice (Supplementary Fig. 2b). These above results are shown in Supplementary Fig. 1b and Fig. 2c and described in our revised manuscript (Page 5, paragraph 2; Page 8, paragraph 2; Page 13, paragraph 1).

2) In Figure 4, the authors should determine whether modulation of Kras and its downstream effectors alters the expression of Nox4, in addition to p22PHOX.

Response: We thank the reviewer for the suggestion. Our published results demonstrated NF- κ B pathway is a novel downstream effector of Kras (Ling et al., Cancer Cell) [8] and others showed that GSK-3 α promotes oncogenic KRAS function in pancreatic cancer via TAK1-TAB stabilization and regulation of noncanonical NF- κ B, suggesting TAK1 is another Kras downstream effector (Bang et al., Cancer Discov. 2013 (6):690-703) [9]. Therefore, we examined the expression NOX4 in the PDAC cells with modulation of Kras and its downstream effectors as suggested. We found that expression of NOX4 was not induced by Dox-regulated mutant Kras in mPDAC/iKras cells. Consistently, NOX4 expression level was not changed in AsPc-1/iTAK1shRNA cells, and Panc-28 or AsPc-1 cells expressing a mutant of I κ B α to inhibit NF- κ B activation. These findings further demonstrate that NOX4 is not regulated though Kras-Tak1/NF- κ B pathway in PDAC cells. These results are shown

in Supplementary Figure 4 as below and described in our revised manuscript (Page 12, paragraph 1).

Supplementary Figure 4 (c) The expression NOX4 was analyzed by immunoblotting in mPDAC/iKras cells with doxycycline-inducible mutant Kras. (d) The expression of NOX4 was analyzed by immunoblotting in AsPc-1/Tak1shRNA inducible cells. (e) The expression NOX4 was analyzed by immunoblotting in wild-type (WT) and IκBα-mutant (Mu) AsPc-1 and Panc-28 cells. β-actin was used as the internal loading control.

3) *The data indicating reduced tumor growth upon inactivation of Nox4 by shRNA could be strengthened by the use of an inducible shRNA, which would allow to determine whether the effect of Nox4 inactivation is due to a defect at the implantation stage or in the maintenance/growth of established tumors.*

Figure S5D

Figure 6f

Supplementary Figure 5 (D) The expression of NOX4 was detected by immunoblotting in AsPc-1/i-shNOX4 cells induced by doxycycline (Dox, 20 ng/ml). The same doxycycline treatment in AsPc-1/Vector control cells caused no significant changes.

Figure 6 (I) Sizes and weights of tumor tissues removed on day 42 from mice injected orthotopically with AsPc-1/i-shNOX4 cells (1×10^6). Dox/on: mice were fed with doxy-containing water from 2 weeks after inoculation, and continued for 4 weeks.

Response: We thank the reviewer for the suggestion. We firstly established the inducible NOX4-knockdown cell line (AsPc-1/i-shNOX4). As shown, addition of doxycycline to the culture medium induced the knockdown of NOX4 protein in AsPc-1/i-shNOX4 cells detected by immunoblotting (Supplementary Fig. 5D). Then, ten mice were orthotopically injected with AsPc-1/i-shNOX4 cells and randomly assigned to two groups (Dox/on and Dox/off). The mice of “Dox/on” group were fed with doxy-containing water from 2 weeks after inoculation. Compared with the control group after 4 weeks of dox treatment, the “Dox/on” group mice showed a significant reduction in tumor burden (Figure 6I). These results indicated that NOX4 play key roles in maintaining the tumor growth in an orthotopic xenograft nude mouse model. These results are shown in Supplementary Figure 5D and Figure 6I as below and described in our revised manuscript (Page 15, paragraph 1).

4) *The increased survival in iKras mice (Figure 6G) is impressive, but would benefit from further analysis. It appears that the survival curve is pushed to the right, yet once the mice start dying the slope is similar as to the control. At the very least histological analysis of the tumor should be performed to determine whether resistance mechanisms have been established. An easy readout would be to determine whether the proliferation index has rebounded. Another possibility is that the tumor grow more slowly, thus reach the critical point later. Understanding a potential mechanism of resistance would be important to assess any therapeutic outcome.*

Response: We apologize for not providing a clear explanation for the last animal experiment. To investigate why DPI treatment just moderately prolonged the survival of the GEMM iKras; p53^{L/+} mice, we analyzed the proliferative index of the dissected tumors as suggested. IHC staining directed against Ki67 revealed a substantially decreased proliferation index in tumors from the DPI treated group after 3 weeks treatment compared to the control group. However, the proliferation index has rebounded in tumors from the DPI treated group after 16 weeks treatment. These findings indicated that the mice tumor may develop resistance to DPI treatment after administration for long time and suggested that highly specific and potent inhibitors of NOX4 deserves further exploration.

Supplementary Figure 6 (c, d) Representative images showing the nuclear expression of the cell proliferation marker Ki 67 and H&E staining in the dissected tumors from the transgenic iKras; p53^{L/+} mice treated with or without DPI for the indicated time.

Diphenyleneiodonium (DPI) is a non-specific inhibitor for NOX4. There might be several potential mechanisms responsible for the DPI resistance. Based on the basis of our published findings (Ju et al Cancer Lett. 2016 May 24), we postulated that the Nampt-mediated NAD salvage pathway may provide a compensatory NAD after NOX4 inhibition, leading to DPI resistance in PDAC. Also, other NOX family members, such as NOX2 or NOX5, may be overexpressed after NOX4 inhibition and play key roles in DPI resistance in PDAC. These possibility should be clarified in our future study and more specific NOX4 inhibitors are under the investigation in our group. These results are shown in Supplementary Figure 6B as below and described in our revised manuscript (page 15, paragraph 1; page 17, paragraph 2).

5) *On a different note, related to the iKras experiment, the authors should comment on the specificity of DPI as a Nox4 inhibitor.*

Response: Actually, diphenyleneiodonium (DPI) is non-specific inhibitor for NOX4. It is a chemical inhibitor of flavoprotein-containing enzymes, including NOX oxidases. Given that inhibition of NOX4 has proven potential for PDAC treatment and DPI just has moderate effects, the development of potent and specific drugs that target NOX4 deserves further exploration. Following the reviewer's suggestion, we have discussed this point in our revised manuscript (page 17, paragraph 2).

Reference

1. Hu, Y., et al., *K-ras(G12V) transformation leads to mitochondrial dysfunction and a metabolic switch from oxidative phosphorylation to glycolysis*. Cell Res, 2012. **22**(2): p. 399-412.
2. Baracca, A., et al., *Mitochondrial Complex I decrease is responsible for bioenergetic dysfunction in K-ras transformed cells*. Biochim Biophys Acta, 2010. **1797**(2): p. 314-23.
3. Weinberg, F., et al., *Mitochondrial metabolism and ROS generation are essential for Kras-mediated tumorigenicity*. Proc Natl Acad Sci U S A, 2010. **107**(19): p. 8788-93.
4. Boland, M.L., A.H. Chourasia, and K.F. Macleod, *Mitochondrial dysfunction in cancer*. Front Oncol, 2014. **3**: p. 292.
5. Chiaradonna, F., et al., *Expression of transforming K-Ras oncogene affects mitochondrial function and*

- morphology in mouse fibroblasts*. *Biochim Biophys Acta*, 2006. **1757**(9-10): p. 1338-56.
6. Warburg, O., *On the origin of cancer cells*. *Science*, 1956. **123**(3191): p. 309-14.
 7. Vander Heiden, M.G., L.C. Cantley, and C.B. Thompson, *Understanding the Warburg effect: the metabolic requirements of cell proliferation*. *Science*, 2009. **324**(5930): p. 1029-33.
 8. Ling, J., et al., *KrasG12D-induced IKK2/beta/NF-kappaB activation by IL-1alpha and p62 feedforward loops is required for development of pancreatic ductal adenocarcinoma*. *Cancer Cell*, 2012. **21**(1): p. 105-20.
 9. Bang, D., et al., *GSK-3alpha promotes oncogenic KRAS function in pancreatic cancer via TAK1-TAB stabilization and regulation of noncanonical NF-kappaB*. *Cancer Discov*. **3**(6): p. 690-703.

REVIEWERS' COMMENTS:

Reviewer #1 (Remarks to the Author):

The authors sufficiently addressed the points raised by the reviewer.

Reviewer #2 (Remarks to the Author):

This revised manuscript includes a substantial amount of new data and has comprehensively addressed my prior concerns. The resulting manuscript should be of broad interest to the field.

Point-by-point replies to the reviewers' comments.

REVIEWERS' COMMENTS:

Reviewer #1 (Remarks to the Author):

The authors sufficiently addressed the points raised by the reviewer.

Response: Thank you very much for reviewing our manuscript. We are very glad that you are satisfied with our responses. We appreciate the reviewer' questions and comments which were very constructive and helpful in strengthening our manuscript.

Reviewer #2 (Remarks to the Author):

This revised manuscript includes a substantial amount of new data and has comprehensively addressed my prior concerns. The resulting manuscript should be of broad interest to the field.

Response: Thank you very much for your review of our manuscript. We appreciate the reviewer' questions and comments which were very constructive and helpful in strengthening our manuscript. We are also very glad to know that the reviewer thought our findings should be of broad interest to the field.